# Topological states between inversion symmetric atomic insulators

## Ana Silva[1] and Jasper van Wezel[2]⋆

**1** Department of Physics of Complex Systems,
Weizmann Institute of Science, Rehovot 76100, Israel
**2** Institute for Theoretical Physics, Institute of Physics,
University of Amsterdam, 1090 GL Amsterdam, The Netherlands

⋆ vanwezel@uva.nl

## Abstract

One of the hallmarks of topological insulators is the correspondence between the value of its bulk topological invariant and the number of topologically protected edge modes observed in a finite-sized sample. This bulk-boundary correspondence has been well-tested for strong topological invariants, and forms the basis for all proposed technological applications of topology. Here, we report that a group of weak topological invariants, which depend only on the symmetries of the atomic lattice, also induces a particular type of bulk-boundary correspondence. It predicts the presence or absence of states localised at the interface between two inversion-symmetric band insulators with trivial values for their strong invariants, based on the space group representation of the bands on either side of the junction. We show that this corresponds with symmetry-based classifications of topological materials. The interface modes are protected by the combination of band topology and symmetry of the interface, and may be used for topological transport and signal manipulation in heterojunction-based devices.



# 1 Introduction

Topological insulators are materials characterised by a bulk topological invariant, which predicts the existence of protected boundary modes localized at the materials' edges [1–3]. This bulk-boundary correspondence has led to proposals for devices exploiting robust electronic, electromagnetic, and mechanical transport properties across a wide range of systems, from quantum materials, to cold atoms, and active classical matter [4–10]. Although there is no general proof for the correspondence between bulk topology and the presence of boundary modes [11], it is commonly accepted that topological insulators with a non-zero value for the so-called strong topological invariant host boundary states that are protected against backscattering [3]. This is true both for systems with broken time-reversal symmetry whose strong invariant arises from the Chern numbers of its bands [2, 12–14], and for time-reversal symmetric systems, where the strong invariant is of the $Z_2$, or Fu-Kane-Mele (FKM), type [15–17]. Both Chern numbers and FKM invariants emerge from a non-zero integrated Berry curvature in the bands of a topological insulator. It has recently been pointed out, however, that a complete classification of topological insulators takes into account the spatial symmetries of the lattice as well as the Berry curvature [18–21]. The relation between the symmetries of the electronic states in an insulator's Bloch bands and the symmetries of the underlying atomic lattice are encoded in band representations, and the numbers of occupied states with given band representations can act as topological invariants in their own right [20–22]. These topological invariants require the presence of lattice symmetries, and are hence of the so-called weak type. Here, the indications 'strong' and 'weak' refer to invariants arising respectively from solely the constraints imposed by intrinsic symmetries, or requiring the presence of lattice symmetries [19, 23]. In this sense, atomic insulators are defined to be insulators with trivial strong invariants. Two such insulators sharing the same lattice symmetry but differing in the number of occupied states with particular band representations, cannot be adiabatically connected without either closing the gap at the Fermi level or breaking the lattice symmetries [19]. There is thus a topological phase transition separating such atomic insulators.

Here, we show that the band representations of two atomic insulators without time-reversal, particle-hole, or chiral symmetries (class A) and with trivial strong topological invariants, can predict the emergence of boundary states localised at the interface between the materials when brought into a heterojunction geometry. This can be seen as a generalised bulk-boundary cor-

respondence, in which the boundary signifies a transition from one material into any topologically distinct medium rather than necessarily connecting to the vacuum. The correspondence naturally complements the classification of topological insulators in terms of band representations, and agrees with its predictions [19]. The fact that the weak invariants in this correspondence are due to band representations, immediately implies that the topological interface states they give rise to will be protected by lattice symmetries only. Nevertheless, the recent progress in the controlled growth and manipulation of heterostructures [24–26], suggests a possible role for topological interface states in mesoscopic devices based on electronic transport properties.

For the sake of being specific, we focus below on two-dimensional systems of spinless fermions, and restrict the discussion to materials with only simple bands and elementary band representations. We detail the analysis of junctions joining two materials with equal space groups, being either p2, p3, or pmm. Based on these examples, we outline the general procedure for identifying topological interface states, and argue that such states generically arise in junctions of inversion symmetric materials.

## 2  Bulk-boundary correspondence

The allowed shapes of electronic wave functions in crystals are determined by the symmetries of its atomic lattice [27]. For atomic insulators with periodic boundary conditions, the real-space wave functions of so-called simple bands (which necessarily are also elementary band representations) [28–31], can be written in the Bloch-like form:

$$\psi^{(\mathbf{w},l)}(\mathbf{k};\mathbf{r}) = \frac{1}{\Omega}\sum_{\mathbf{R}_n} e^{i\mathbf{k}.\mathbf{R}_n} a^{(\mathbf{w},l)}(\mathbf{r}-\mathbf{R}_n). \tag{1}$$

Here, $\mathbf{R}_n$ are the centres of the unit cells, while $\mathbf{w}$ is a Wyckoff position, i.e. a point inside the unit cell that is mapped onto itself (modulo lattice translations) by the point group of the crystal or one of its subgroups. The Wannier functions $a^{(\mathbf{w},l)}$ are symmetry-appropriate basis functions centered at $\mathbf{w}$, which means that under the point group operations leaving $\mathbf{w}$ invariant, these functions transform as the irreducible representation labeled by $l$. Notice that the index $(\mathbf{w},l)$ applies for any value of $\mathbf{k}$, and thus determines the real-space symmetry of the entire Bloch band [32]. These real-space labels are an alternative to labelling bands by a set of irreducible representations at high-symmetry points in momentum space, and for the p2 and pmm-symmetric atomic insulators considered here, there is a one-to-one relation between the two conventions [33–35].

In crystals with open boundary conditions, bulk states become standing waves built from linear combinations of Bloch states with opposing momenta, with small deformations that allow them to smoothly connect to decaying vacuum solutions outside the crystal or to different standing waves in a neighbouring crystal [36–38]. For standing waves built from Bloch states at high-symmetry momenta, lattice symmetry alone may force either the real-space wave function or its derivative to be zero at Wyckoff positions, making it impossible to connect to the outside if the material terminates at such a position. The only way to overcome this obstruction, is for the Bloch momentum value to acquire an imaginary component, corresponding to an exponential decay inside the bulk of the crystal. In other words, whenever the existence of bulk states is prevented by lattice symmetry, an exponentially localised edge state arises [39]. Moreover, because the bulk standing waves are constructed from linear combinations of Bloch states, their matching conditions at the crystal's boundary can be written entirely in terms of Bloch state properties. There is thus a direct correspondence between properties of bulk states calculated with periodic boundary conditions, and the existence of edge or interface states in

corresponding materials with boundaries.

The obstruction to constructing bulk states in strong topological insulators is signalled by the presence of non-zero net Berry curvature [40,41]. Atomic insulators have zero net Berry curvature, but they do carry symmetry-based topological invariants whose values may vary between different insulators with the same lattice symmetry [19–21]. An obstruction to constructing bulk states connecting two such insulators may then be formulated in terms of the symmetry labels $(\mathbf{w}, l)$ of occupied bands. In the presence of an obstruction, boundary states will form at the interface separating the two insulators. For one-dimensional crystals with inversion symmetry, this bulk-boundary correspondence has been explicitly constructed [39, 42]. Here, we extend these results to two (and higher) dimensions, and show how they can be used to predict the presence of localised states at the interface joining two atomic insulators.

## 3 Interface states

For electronic states to smoothly connect across an interface, the wave functions and their derivatives need to match on either side [36,43]. It has been shown that this smooth matching of standing waves in an open system is possible only if a similar smooth connection can be made using the Bloch waves of the corresponding periodic bulk [36–38]. This can be seen as an incarnation of the bulk-boundary correspondence, which allows us to predict the feasibility of matching wave functions in an open system by studying the shape of Bloch waves in the periodic setup. Moreover, if no obstruction for joining together bulk states exists, localised edge state solutions do not arise [39]. To find topological edge states, it thus suffices to check whether Bloch states with complex momentum values can be smoothly connected across an interface. This is most conveniently done using the logarithmic derivative, defined as:

$$\rho^{(\mathbf{w},l)}(\mathbf{k}, \mathbf{r}) = \frac{\mathbf{n} \cdot \boldsymbol{\nabla} \psi^{(\mathbf{w},l)}(\mathbf{k}, \mathbf{r})}{\psi^{(\mathbf{w},l)}(\mathbf{k}, \mathbf{r})} \,. \tag{2}$$

Here, $\mathbf{n}$ is a vector normal to the boundary. The logarithmic derivative is invariant under both gauge transformations (changing the Wannier basis) and translations by lattice vectors. Moreover, for any space group that includes inversion symmetry and allows for real-valued Wannier functions, such as p2 or pmm, $\rho(\mathbf{k}, \mathbf{r})$ can be shown to be purely imaginary for any real value of momentum, and purely real whenever the real part of $\mathbf{k}$ is a high-symmetry point in the Brillouin zone [42] (see Appendix B.3 for details).

This is particularly convenient in the search for boundary states, which arise only at high-symmetry momenta. That is, solutions to Schrödinger's equation with real $\mathbf{k}$ exist for any momentum value along the bulk of any band. At high-symmetry points, however, the lattice symmetry forces either the wave function or its derivative to vanish, and these conditions may pose an obstruction to creating smoothly connected states across the boundary. In that case, there will be an edge mode with $\mathbf{k}_{\text{edge}} = \mathbf{k}_{\text{R}} + i \mathbf{k}_{\text{I}}$, where $\mathbf{k}_{\text{R}}$ is the high-symmetry momentum value. Since the logarithmic derivative for $\mathbf{k}_{\text{edge}}$ is purely real, an edge mode can only arise if the signs of $\rho(\mathbf{k}_{\text{edge}}, \mathbf{r})$ agree on either side of the junction. Here, $\mathbf{r}$ denotes the location of the edge, which should coincide with a high-symmetry point of the lattice. In fact, the matching of signs has been shown to be both a necessary and sufficient condition [39].

For inversion-symmetric crystals, the lattice symmetry causes a zero in either the Bloch state or its derivative at any high-symmetry momentum, forcing $\rho(\mathbf{k}, \mathbf{r})$ to go to either zero or infinity [42]. Other, accidental zeroes or infinities always come in pairs, and may be ignored without loss of generality. Given the band label $(\mathbf{w}, l)$ and a location $\mathbf{r}$ along the crystal boundary (chosen to be a Wyckoff position), the zeroes and infinities of $\rho$ at high-symmetry momenta are entirely fixed by the crystal symmetry (as shown in Appendix A for p2, pmm,

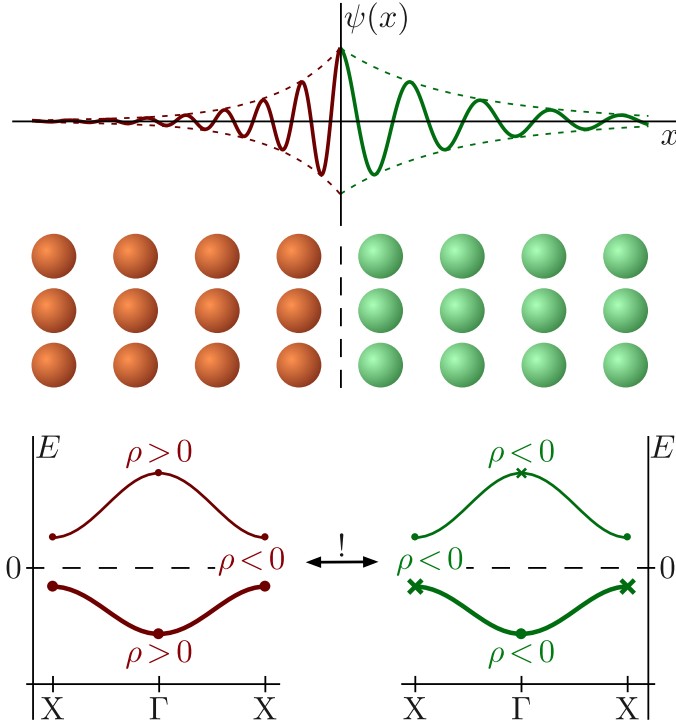

Figure 1: **Topological mode at the interface of two atomic insulators.** Exponentially localised states (top) may arise at the interface separating two inversion-symmetric atomic insulators (middle), if their individual band structures present a topological obstruction for bulk states to connect across the interface (bottom). The obstruction is diagnosed by the real-space symmetry labels of the occupied bands (thick lines), which uniquely determine the zeroes of the Bloch wave function (solid dots) and its derivative (crosses) at high-symmetry points in the Brillouin zone. These in turn yield the sign of the logarithmic derivative $\rho$ in any of the bulk energy gaps. Whenever the sign is the same for gaps at equal energies at either side of the interface, a localised topological interface state emerges.

and p3). Together with the fact that Bloch functions can be analytically continued to complex momentum values [44–46], these zeroes and infinities in turn fix the sign of the logarithmic derivative for any putative boundary state [42].

Consider for example the band with zeroes at both $\Gamma$ and $X$, and assume that $\rho$ starts off positive for boundary states near $\Gamma$, as shown schematically in the bottom left panel of Fig. 1 That is, for small $\epsilon$ a power series expansion of the logarithmic derivative yields $\rho(\Gamma-i\epsilon) \sim \epsilon$. This can be analytically continued to propagating bulk modes near $\Gamma$ by writing it as $\rho(\Gamma-z) \sim -iz$, which implies $\rho(\Gamma+\epsilon) \sim i\epsilon$. When traversing the band from $\Gamma$ to $X$, the logarithmic derivative has to stay purely imaginary. Moreover, it cannot pass through the origin since there are no zeroes or infinities outside of the high-symmetry momenta. We thus find that $\rho(X-\epsilon) \sim i\epsilon$ as well. This can again be analytically continued toward boundary states near $X$ by first writing $\rho(X-z) \sim iz$, and thus $\rho(X-i\epsilon) \sim -\epsilon < 0$. A band connecting two zeroes thus causes a change in sign of the logarithmic derivative when going from one band gap to the next (Fig. 1, bottom left panel). If instead there had been an infinity at $X$, we would have found $\rho(X-\epsilon) \sim i/\epsilon$, and thus $\rho(X-i\epsilon) \sim 1/\epsilon > 0$ (Fig. 1, bottom right). Continuing in this way, we find that $\rho$ changes sign across a band with either two zeroes of two infinities, and stays fixed otherwise. All signs of $\rho$ are thus fixed by the lattice symmetry once the band

labels $(\mathbf{w}, l)$ are given.

In a heterojunction of two atomic insulators, a boundary state localised at the interface will arise whenever the two materials have a band gap over the same range of energies, and the logarithmic derivatives on either side of the junction have the same sign for some complex $\mathbf{k} = \mathbf{k}_R + i\mathbf{k}_I$, with $\mathbf{k}_R$ a high-symmetry point. Since the symmetry labels of the bands in p2 and pmm-symmetric crystals fully determine the sign changes of $\rho$ on either side of the junction, they also predict in which band gaps interface states will form, as shown schematically in the example of Fig. 1.

Notice that this argument requires both inversion symmetry and the existence of real-valued Wannier states to relate the effect of symmetry operations to that of complex conjugation. The three-fold rotation in p3, for example, can have eigenvalues $e^{\pm i2\pi/3}$, which cannot be realised in any real-valued Wannier state. For bands characterised by such eigenvalues, the real-space symmetry operations still impose constraints on the logarithmic derivative, for example relating the signs along symmetry-related edges (as shown in Appendix A.3 for p3), but in general, these do not sufficiently restrict $\rho$ to predict the presence or absence of interface states. Similarly, the absence of inversion symmetry in p3 also prevents the prediction of interface states for its real-valued Wannier states.

## 4  Topology of interface states

Interface states between crystals with different symmetry labels occurring at the Fermi level ($E_F$) are expected to be topological in nature, based on the classification of topological insulators with crystalline symmetry [19]. Since topological phase transitions by definition involve either the closing of the band gap at the Fermi level or a change of lattice symmetry, smooth deformations of the Hamiltonian that do neither of these things should have no effect on the interface states. To see that this is indeed the case for the interface states discussed here, notice that the analysis predicting their presence at $E_F$ depends only on the band symmetry labels on either side of the junction [1]. The only deformations of the Hamiltonian that can affect the interface states then, are changes in the symmetry labels. These are caused by band inversions at high-symmetry points in momentum space.

In atomic insulators it is always possible to induce the momentum space representations at high-symmetry points from the real-space band labels [34, 47]. The former are more general, in the sense that they describe strong topological insulators with non-zero Chern numbers as well as atomic insulators, whereas the real-space band symmetry labels can be applied only to atomic insulators [20, 32, 41]. For crystals with p2 or pmm symmetry, the mapping from real space to momentum space labels is especially straightforward. For these groups, the lattice symmetry causes either the Bloch state or its derivative to go to zero at all high symmetry momenta, depending on the band symmetry label. At the same time, a Bloch state can only have zero slope if it is even under reflection, and have zero value (but not zero slope) if it is odd. The zeros and infinities of the logarithmic derivative $\rho$ thus indicate how the Bloch state transforms under inversion at high-symmetry points, or under mirror operations along high-symmetry lines. Together with consistency requirements throughout the Brillouin zone, this completely fixes the momentum-space representation (see Appendix C for details).

A band inversion at a high-symmetry point can be regarded as the exchange of momentum-space representations between two bands at that point. Because for p2 and pmm the representations entirely determine the value of $\rho$, the band inversion can equivalently be described as

---

[1] Strictly speaking, it also depends on the sign of $\rho$ for energies below the lowest band, which is not determined by the symmetry labels. This sign, however, is fixed by the physical requirement that there are no edge states below the lowest occupied band.

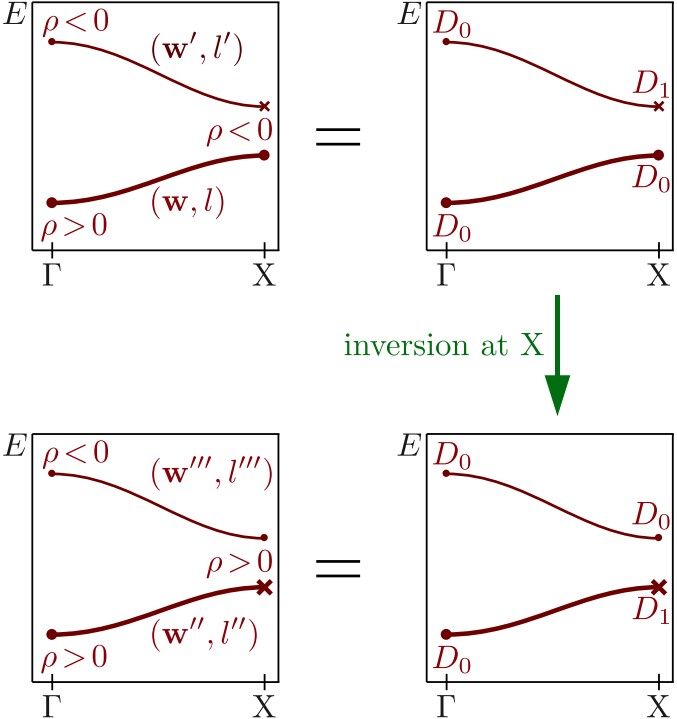

Figure 2: **The effect of a band inversion.** In inversion-symmetric atomic insulators, the real-space symmetry labels $(\mathbf{w}, l)$ that determine the sign of the logarithmic derivative in bulk energy gaps (top left), also fix the representations of the space group symmetries relevant to high-symmetry points in the Brillouin zone (top right). A band inversion implies the exchange of representations across an energy gap at one or several of the high-symmetry points (bottom right). This may result in the logarithmic derivative $\rho$ changing sign in that gap, and in the symmetry-labels of adjacent bands being altered, but it does not affect the signs of $\rho$ in any other bulk energy gaps (bottom left).

an exchange of the zeroes or infinities of $\rho$ at the high-symmetry point (see Fig. 2). As shown before, the sign of $\rho$ for edge states associated with a particular high-symmetry point $\mathbf{k}$ can be determined based on the sign of $\rho$ in a lower-energy band gap associated with a different point $\mathbf{k}'$ (recall Fig. 1). If the line connecting $\mathbf{k}'$ and $\mathbf{k}$ has either two zeroes or two infinities at its end points, the sign of $\rho$ changes from one gap to the next. For one zero and one infinity, the signs are equal. If we now consider a succession of two bands, as in Fig. 2, interchanging the zeroes or infinities in the middle can be easily seen not to affect the relation between the sign of $\rho$ below the lowest and above the highest band. Either two bands with no sign flips transform into two bands that both flip the sign of $\rho$ (or the other way around), or a band with a sign flip changes places with a band without. As long as only occupied bands undergo band inversions, the relation between the sign of $\rho$ in the topmost gap, at the Fermi level, and the sign at the bottom, below all occupied bands (which is fixed) thus stays constant. That is, the sign of $\rho$ at $E_F$, and hence the presence of interface states, can only be changed in a topological phase transition, involving the closing of the gap between valence and conductance bands or a change in lattice symmetry.

The correspondence of the topology of interface states predicted by real-space band symmetry labels and the momentum-space classification of crystalline topological insulators can be made by realising that band inversions which exchange zeroes and infinities of $\rho$ at high-

symmetry momenta are accompanied by the exchange of representations at those points. This does not change the total number of bands in any given representation at a high-symmetry point, which was found to be the most general topological invariant for crystalline topological insulators [19–21]. The band inversions that do not affect the sign of $\rho$ at $E_F$, and thus leave the edge states fixed, are also precisely the transformations of the Hamiltonian that do not change its topological classification.

# 5 Examples

To illustrate the general results of the previous sections, we will consider two explicit examples in which interface states between two atomic insulators are realised according to the prediction based on symmetry labels of the bulk Hamiltonians. First, we build a two-dimensional material from a stack of one-dimensional Su-Schrieffer-Heeger (SSH) models [48,49], described by the Hamiltonian:

$$
\begin{aligned}
\hat{H} = \sum_{x,y} & v\left(\hat{c}^\dagger_{x,y}\,\hat{d}_{x,y} + \hat{d}^\dagger_{x,y}\,\hat{c}_{x,y}\right) + w\left(\hat{d}^\dagger_{x,y}\,\hat{c}_{x+1,y} + \hat{c}^\dagger_{x+1,y}\,\hat{d}_{x,y}\right) \\
& + t\left(\hat{c}^\dagger_{x,y}\,\hat{c}_{x,y+1} + \hat{c}^\dagger_{x,y+1}\,\hat{c}_{x,y} + \hat{d}^\dagger_{x,y}\,\hat{d}_{x,y+1} + \hat{d}^\dagger_{x,y+1}\,\hat{d}_{x,y}\right) \\
& + t'\left(\hat{c}^\dagger_{x,y}\,\hat{d}_{x,y+1} + \hat{d}^\dagger_{x,y+1}\,\hat{c}_{x,y}\right).
\end{aligned}
\tag{3}
$$

Here, $x$ labels the unit cell position along each chain while $y$ labels the chains. The unit cell at $(x, y)$ contains two sites, with corresponding electron creation operators $\hat{c}^\dagger_{x,y}$ and $\hat{d}^\dagger_{x,y}$. The $t'$ hopping integral breaks all mirror symmetries, leaving a lattice with $p_2$ symmetry.

We consider an interface between two materials that are both described by the Hamiltonian $\hat{H}$, but with one having $v \gg w > t, t'$ and the other having $w \gg v > t, t'$, as shown in the top left panel of Fig. 3. Using periodic boundary conditions, the bulk band structures can be calculated for the isolated materials on either side of the interface. These are shown in the bottom left of Fig. 3, along with the band representations at the high-symmetry points in the Brillouin zone. Because the only point group in the lattice is a two-fold rotation, there are only two possible representations, being either even (indicated by a plus sign) or odd (indicated by a minus). As shown in Appendix C, the four representation labels of each band together correspond to a single real-space symmetry label $(\mathbf{w}, l)$ with $\mathbf{w} \in \{\mathbf{w}_a, \mathbf{w}_b\}$ and $l \in \{0, 1\}$, which is indicated alongside each band.

Upon specifying a particular boundary for each of the two materials, as indicated in the top left panel of Fig. 3, the real-space symmetry labels imply zeroes for either the Bloch wave function at a high-symmetry momentum value, or its derivative (see Appendix A). These are indicated in the band structure by solid dots and crosses respectively. In turn, the zeroes or divergencies determine whether or not the logarithmic derivative $\rho$ differs in sign between the gaps above and below each band, as shown in Appendix B. Starting from opposite signs of $\rho$ below the lowest band, the logarithmic derivative then ends up having equal sign in the band gap between the first and second band, as indicated in Fig. 3. We thus expect an interface state to form in that gap.

The stack of SSH chains has the advantage of being an intuitive model, in which the interface state can be understood as a stack of edge states in the individual one-dimensional chains which survive in two dimensions despite the absence of any type of strong invariant. However, the stack of SSH chains has a chiral symmetry in addition to the two-fold rotational symmetry of its atomic lattice, as seen from the fact that the operator $\hat{c}^\dagger_{x,y}\hat{c}_{x,y} - \hat{d}^\dagger_{x,y}\hat{d}_{x,y}$ anti-commutes with the Hamiltonian. This chiral symmetry has no influence on the analysis of band labels and the corresponding prediction of localised interface states. For completeness, however, we

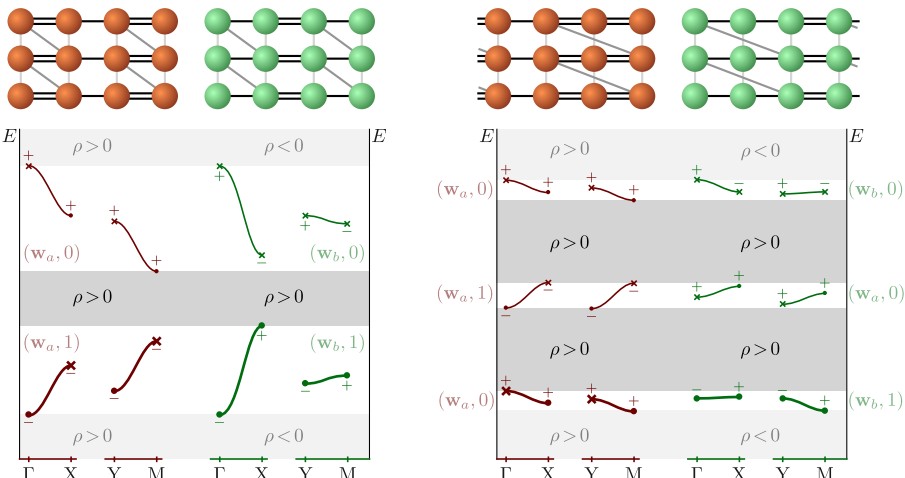

Figure 3: **Examples of interfaces.** Two examples of interfacing inversion-symmetric atomic insulators with identical lattice symmetries but different numbers of occupied states for their allowed band representations. The top row schematically depicts the models, consisting of stacks of SSH chains (left) or stacks of period-three CDWs (right). In each case, the strengths of the inter-unit cell and intra-unit cell hopping integrals along the chain direction are exchanged across the interface. The bottom row shows high-symmetry cuts of the band structures for each of the four two-dimensional models. For each band, the representations of the states at high-symmetry points in the Brillouin zone are indicated by plus (even under two-fold rotations) or minus (odd) signs. These representations correspond one-to-one to the real-space symmetry labels $(\mathbf{w}, l)$ that are also indicated for each band. All of these are calculated in the presence of periodic boundary conditions and represent bulk properties. Upon introducing a boundary or interface as indicated in the top row, the real-space symmetry labels uniquely determine the zeroes of the Bloch wave function (solid dots) and its derivative (crosses) at high-symmetry points in the Brillouin zone. These in turn yield the signs of the logarithmic derivative in bulk energy gaps. Whenever the sign is the same for gaps at equal energies at either side of the interface (indicated by a darker shading), a localised topological interface state is predicted to emerge. In this plot, we used the dimensionless parameter values $v = 20$, $w = 6$, $t = 1$, and $t' = 5$ for the leftmost configuration of SSH chains and $v = 30$, $w = 5$, $t = 1$, and $t' = 1$ for the leftmost configuration of CDWs.

also consider a second, less intuitive, model with $p_2$ lattice symmetry, which does not have additional chiral symmetries. This second model consists of a stack of period-three charge density waves (CDWs) [50, 51], as depicted in the top right panel of Fig. 3. The corresponding Hamiltonian is given by:

$$\hat{H} = \sum_{x,y} v \left( \hat{c}_{x,y}^\dagger \hat{d}_{x,y} + \hat{d}_{x,y}^\dagger \hat{f}_{x,y} + \text{H.c.} \right) + w \left( \hat{f}_{x,y}^\dagger \hat{c}_{x+1,y} + \text{H.c.} \right),$$

$$+ t \left( \hat{c}_{x,y}^\dagger \hat{c}_{x,y+1} + \hat{d}_{x,y}^\dagger \hat{d}_{x,y+1} + \hat{f}_{x,y}^\dagger \hat{f}_{x,y+1} + \text{H.c.} \right) + t' \left( \hat{c}_{x,y}^\dagger \hat{f}_{x,y+1} + \text{H.c.} \right). \tag{4}$$

Here, H.c. indicates the Hermitian conjugate.

We again consider two materials with the same form for their Hamiltonians, one having $v \gg w \gg t, t'$ and the other having $w \gg v \gg t, t'$. The bottom right panel of Fig. 3 shows the bulk band structures, along with the band representations at high-symmetry points in the Brillouin zone as well as the corresponding real-space symmetry labels. Also indicated are the

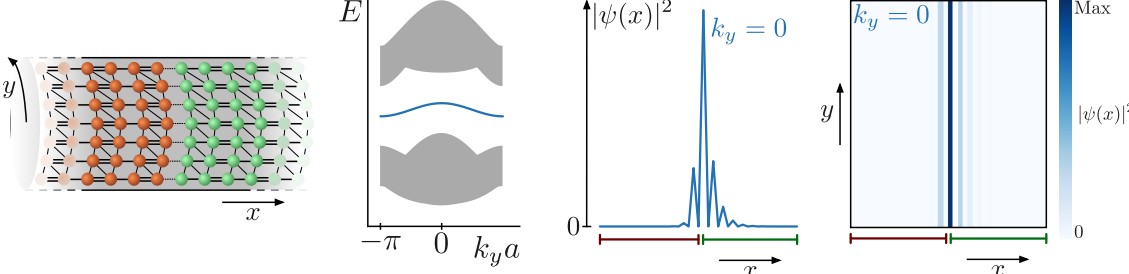

Figure 4: **Topological interface state.** An example of a topological interface state between two atomic insulators, predicted on the basis of bulk symmetry labels. The real-space configuration consists of two-dimensional materials with periodic boundary conditions in one direction, meeting at a shared interface (leftmost panel). The parameter values are the same as in Fig. 3, with an extra bond of strength $(v + w)/2$ connecting neighbouring atoms directly across the interface. The resulting spectrum (second panel) contains two bulk bands (grey shading) and one topological in-gap band. The real space wave function for the in-gap state at $k_y = 0$ is shown at $y = 0$ in the third panel, and as a heat map for all $(x, y)$ in the rightmost panel. The topological state is seen to be exponentially localised at the interface.

zeroes and divergencies of the Bloch functions for the specific choice of interface shown in the top right panel, and the corresponding sign structure of the logarithmic derivatives in the bulk band gaps. Just as for the stack of SSH chains, we predict to find localised interface states. For the stack of CDWs, they arise in both gaps surrounded by bulk bands.

To check the predicted presence of interface states, the two SSH-stack Hamiltonians with different values of $v$ and $w$ are connected as shown in Fig. 3. The sites directly bordering the interface are connected by a horizontal hopping of strength $(v + w)/2$. We then introduce periodic boundary conditions along the $y$ direction to end up with a cylindric geometry, as depicted in Fig. 4. The momentum in the $y$ direction is then a good quantum number, and the spectrum for the entire cylinder is shown in the second panel of Fig. 4. As expected, it has a state in the band gap, well separated from the bulk bands. Plotting the wave function for the $k_y = 0$ in-gap state in real space, as shown in the right two panels of Fig. 4, clearly shows it to be an exponentially localised interface state, as predicted by the real-space symmetry labels of the two bulk materials at either side of the interface. For a cylinder with stacks of CDWs on either side of the interface, we obtain similar interface states.

The band of interface states does not connect the upper and lower bulk bands. This is to be expected, since we consider an interface between atomic insulators with trivial strong invariants for all bands. Although this feature does not affect the existence of interface states, it does have an impact on their robustness. The interface band not being connected to any of the bulk bands in principle allows it to be perturbed and moved in energy by the addition of impurities or impurity potentials at the interface. Again, this is not unexpected for weak topological states that are protected only by lattice symmetries [15], and is also the case for example for the well-known topological end states of the one-dimensional SSH chain (if the impurity breaks the chiral symmetry) and the period-three CDW [42]. To see how much the interface states are influenced by impurities in practice, we show in Fig. 5 the interface in a cylinder geometry, but with a single impurity potential located at the interface, indicated with a red cross. We again show only the results for the SSH-stack model, but obtain equivalent results for the CDW-stack model. Since the impurity breaks translational symmetry along the $y$ direction, the spectrum is shown as a single tower of eigenenergies in the second panel of

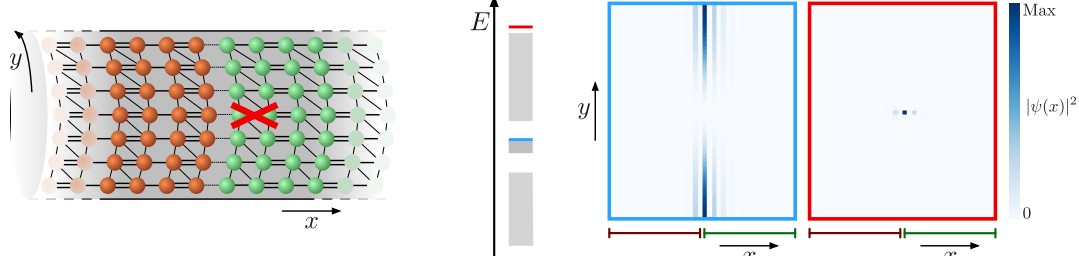

Figure 5: **Topological interface state with an impurity.** An example of the effect a single localised impurity potential (indicated by a red cross in the leftmost panel) has on the topological interface state between two atomic insulators. The real-space configuration consists of two-dimensional materials with periodic boundary conditions in one direction, meeting at a shared interface (leftmost panel). The parameter values are the same as in Figs. 3 and 4, with an additional very strong impurity potential $V = 30$ at the central site. The resulting spectrum (second panel) contains two bulk bands (lowest grey band, and third grey band from the bottom), one topological in-gap band (second grey band from the bottom), and one isolated impurity state (red line). The real space wave function for the highest-energy in-gap state (blue line, corresponding to $k_y = 0$ if the impurity is absent) is shown in the third panel with the blue outline, and that of the impurity state in the rightmost panel with the red outline. The topological state is seen to span the system in spite of the very strong impurity potential at the center.

Fig. 5. Even for a very strong impurity potential, exceeding the bulk bandwidth, the topological impurity band can be seen to survive within the bulk band gap, while a single localised mode at the impurity site is formed at very high energies. The real space wave functions of the topological states are suppressed at the location of the impurity (as shown in the third panel), but still span the entire interface. Even very strong impurities therefore do not necessarily present any obstruction to employing topological interface states in practice.

## 6 Discussion

We have shown that crystalline topological insulators may harbour topologically protected edge states, even in the atomic limit. These edge states are not associated with any Berry curvature or intrinsic symmetry, but rather rely on the symmetries of the atomic lattice itself. We have shown that these states arise naturally at the interface between two-dimensional atomic insulators with inversion symmetry and real-valued Wannier states, which may be realised in heterojunction architectures. The inversion symmetry protecting the interface states can be realised in two dimensions either as a two-fold rotation or as the product of two mirror operations. In higher dimensions, the same arguments are expected to apply in the presence of any symmetries relating states at $\mathbf{k}$ and $-\mathbf{k}$. These could be three mirror planes, combinations of mirror planes and rotation axes, or a three-dimensional inversion centre. The interface states are topologically robust in the sense that they cannot be removed by smooth deformations of the bulk Hamiltonian that do not close the conductance gap or change the symmetry of the atomic lattice.

The existence of interface states does in general depend sensitively on the terminations of the crystals on either side of the junction, as is the case for any type of weak topology protected by lattice symmetry [15,17,42]. This sensitivity may disappear, however, in a special

type of heterostructure, with the same atomic lattice on either side. Such interfaces may arise for example when inhomogeneous doping or external fields are applied to a single crystal, causing phase separation without affecting the atomic structure. In Appendix B.5, it is shown explicitly that in such cases, with p2 or pmm symmetry, shifting the location of the interface always affects the sign of $\rho$ on both sides of the junction in the same way, leaving their product invariant. For this special type of heterojunction, the existence of interface states is robust not only to band inversions, but also to changes and defects in the interface geometry.

**Author contributions**  All authors contributed to all stages of the research.

# A  Symmetry-imposed zeroes of Bloch wave functions and their derivatives

## A.1  $p_{mm}$ symmetry

In a crystal with $p_{mm}$ space group symmetry, there are four inequivalent Wyckoff positions [52]. Each of these are left invariant by the symmetry operations of the point group $D_2$, whose characters are given in table 1 [27]. The non-trivial group operations are reflections $(x, y) \to (-x, y)$ and $(x, y) \to (x, -y)$, written $M_x$ and $M_y$, and their product, which is equivalent to a 180° rotations around an axis perpendicular to the plane and is therefore denoted $C_2$. For each of the four Wyckoff positions $\mathbf{w}$, we can define a phase factor associated with the symmetry operation $\alpha$ as $\mathbf{R}_{\mathbf{w}}^{\alpha} = \mathbf{w} - \alpha\mathbf{w}$ [33]. The phase factors for the non-trivial operations in the $D_2$ point group are listed in table 2.

For each of the non-trivial point group operations $\alpha$, the wave function $\psi^{(\mathbf{w},l)}$, built from Wannier functions centered at Wyckoff position $w$ and transforming as representation $D_l$ of the point group, is constrained by the following relations [33]:

$$\alpha\,\psi^{(\mathbf{w},l)}(\mathbf{k};\mathbf{r}) = \psi^{(\mathbf{w},l)}(\alpha^{-1}\mathbf{k};\alpha^{-1}\mathbf{r}),$$
$$= e^{i\mathbf{k}\cdot\mathbf{R}_{\mathbf{w}}^{\alpha}}\,D_l(\alpha)\,\psi^{(\mathbf{w},l)}(\mathbf{k};\mathbf{r}). \tag{5}$$

Taking both $\mathbf{k}$ and $\mathbf{r}$ to be invariant under the operation of $\alpha$, these conditions can be met only if either $e^{i\mathbf{k}\cdot\mathbf{R}_{\mathbf{w}}^{\alpha}}D_l(\alpha) = 1$, or the wave function is zero.

For example, the symmetry operation $M_x$ leaves invariant any $\mathbf{k}$-point on the lines $\mathbf{\Gamma Y}$ and $\mathbf{XM}$ (up to translations by a reciprocal lattice vector), and any real-space point on lines $x = 0$ and $x = a/2$ (modulo translations by a lattice vector). For a crystal terminating at a boundary with $x = 0$, we find $\mathbf{R}_{\mathbf{w}}^{M_x} = 0$, so that any band transforming as either $D_1$ or $D_3$ will be forced to have its wave function go to zero on the lines $\mathbf{\Gamma Y}$ and $\mathbf{XM}$ in the Brillouin zone. The symmetry is thus particularly important for crystals terminating at a boundary with fixed $x$ coordinate, in which symmetry imposes the suppression of the wave function and hence the appearance of an edge state along an entire boundary.

Besides imposing zeroes on the Bloch wave function, the symmetry may also impose zeroes on the directional derivatives of the Bloch wave function. For example, if a Bloch wave function is even under the reflection $(x, y) \to (-x, y)$, then its normal derivative along $x$ is necessarily odd under the same reflections. That is, if the wave function transforms as $D_0$ or $D_2$, its normal derivative along $x$ will transform as respectively $D_3$ or $D_1$, and vice versa. Equation (5) then imposes zeroes on the normal derivatives in the same way as it does for the Bloch wave function itself.

Tables 3 through 6 list the symmetry-imposed zeroes in both the Bloch wave function and its normal derivatives for all possible combinations of high-symmetry positions in real and reciprocal space.

Table 1: Character table for the point group $D_2$, relevant at any of the Wyckoff positions in a $p_{mm}$-symmetric atomic insulator. $E$ labels the identity operation.

|       | $E$ | $M_x$ | $M_y$ | $C_2$ |
|-------|-----|-------|-------|-------|
| $D_0$ | 1   | 1     | 1     | 1     |
| $D_1$ | 1   | -1    | -1    | 1     |
| $D_2$ | 1   | 1     | -1    | -1    |
| $D_3$ | 1   | -1    | 1     | -1    |

Table 2: The phase factor associated with the non-trivial $D_2$ point group operations for the four Wyckoff positions in a $p_{mm}$-symmetric atomic insulator.

| $\mathbf{w}$ | $\mathbf{R}_{\mathbf{w}}^{M_x}$ | $\mathbf{R}_{\mathbf{w}}^{M_y}$ | $\mathbf{R}_{\mathbf{w}}^{C_2}$ |
|---|---|---|---|
| $\mathbf{w}_a = (0,0)$ | $\mathbf{0}$ | $\mathbf{0}$ | $\mathbf{0}$ |
| $\mathbf{w}_b = (\frac{a}{2},0)$ | $a\mathbf{e}_x$ | $\mathbf{0}$ | $a\mathbf{e}_x$ |
| $\mathbf{w}_c = (0,\frac{a}{2})$ | $\mathbf{0}$ | $a\mathbf{e}_y$ | $a\mathbf{e}_y$ |
| $\mathbf{w}_d = (\frac{a}{2},\frac{a}{2})$ | $a\mathbf{e}_x$ | $a\mathbf{e}_y$ | $a\mathbf{e}_x + a\mathbf{e}_y$ |

## A.2  $p_2$ symmetry

Crystals with $p_2$ space group symmetry, again have four inequivalent Wyckoff positions, $(0,0)$, $(a/2,0)$, $(0,a/2)$, and $(a/2,a/2)$ [52], and four high symmetry points in the Brillouin Zone, $\mathbf{\Gamma}$, $\mathbf{X}$, $\mathbf{Y}$, and $\mathbf{M}$. They are all left invariant by the elements of the point group $C_2$, whose character table [27] and phase factors $e^{i\mathbf{k}\cdot\mathbf{R}_{\mathbf{w}}^{C_2}}$ are provided in table 7. Applying equation (5) again yields symmetry-enforced zeroes, listed in table 8.

## A.3  $p_3$ symmetry

In a $p_3$-symmetric crystal, there are three inequivalent Wyckoff positions, at $(0,0)$, $(2a/3,a/3)$, and $(a/3,2a/3)$ [52], written as $(x,y) \equiv x\,\hat{\mathbf{e}}_1 + y\,\hat{\mathbf{e}}_2$ in terms of the unit vectors $\hat{\mathbf{e}}_1 = -\frac{1}{2}\hat{\mathbf{e}}_x + \frac{\sqrt{3}}{2}\hat{\mathbf{e}}_y$ and $\hat{\mathbf{e}}_2 = \hat{\mathbf{e}}_x$. These positions are left invariant by the symmetry operations of the point group $C_3$, with the character table [27] and phase factors presented in (9). The symmetry operations include 120° and 240° rotations, denoted as $C_3$ and $C_3^2$ respectively.

   For representations of $p_3$-symmetry which do not allow real-valued Wannier functions, the behavior of the logarithmic derivative is less constrained. As noticed before, the effects of the space group symmetry on a crystal edge are most pronounced along edges that respect one of the non-trivial symmetry operations. Since it is not possible for any individual edge to be left invariant under three-fold rotations, we consider a crystal shaped like an equilateral triangle (see figure 6), whose sides are mapped onto one another under $C_3$ rotations. The corners are at positions $-Na\,\hat{\mathbf{e}}_2$, $Na\,(\hat{\mathbf{e}}_1 + \hat{\mathbf{e}}_2)$, and $-Na\,\hat{\mathbf{e}}_1$, with $N$ an integer, while the normal vectors to the boundaries are given by $\mathbf{n}_1 = \hat{\mathbf{e}}_1$, $\mathbf{n}_2 = \hat{\mathbf{e}}_2$ and $\mathbf{n}_3 = -(\hat{\mathbf{e}}_1 + \hat{\mathbf{e}}_2)$.

   The normal derivatives at the three boundaries can be expressed in terms of just two functions:

$$
\begin{aligned}
B_1^{(\mathbf{w},l+1)} &= (e^{i\pi/3}\hat{\mathbf{e}}_1 + \hat{\mathbf{e}}_2) \cdot \nabla \psi^{(\mathbf{w},l)}, \\
B_2^{(\mathbf{w},l-1)} &= (e^{-i\pi/3}\hat{\mathbf{e}}_1 + \hat{\mathbf{e}}_2) \cdot \nabla \psi^{(\mathbf{w},l)}.
\end{aligned}
\tag{6}
$$



Table 3: Zeroes imposed by $p_{mm}$-symmetry on either the Bloch wave function or its normal derivative along $\hat{\mathbf{e}}_x$. The real-space positions are fixed at $\mathbf{r} = (0, y)$, for arbitrary values of the $y$ coordinate.

| $(w,l)$ , $\mathbf{r}$ | $\mathbf{k} = \Gamma$ | X | Y | M |
|---|---|---|---|---|
| $(\mathbf{w}_a,0)$ , $(0,y)$ | $\partial_x \psi = 0$ | $\partial_x \psi = 0$ | $\partial_x \psi = 0$ | $\partial_x \psi = 0$ |
| $(\mathbf{w}_a,1)$ , $(0,y)$ | $\psi = 0$ | $\psi = 0$ | $\psi = 0$ | $\psi = 0$ |
| $(\mathbf{w}_a,2)$ , $(0,y)$ | $\partial_x \psi = 0$ | $\partial_x \psi = 0$ | $\partial_x \psi = 0$ | $\partial_x \psi = 0$ |
| $(\mathbf{w}_a,3)$ , $(0,y)$ | $\psi = 0$ | $\psi = 0$ | $\psi = 0$ | $\psi = 0$ |
| $(\mathbf{w}_b,0)$ , $(0,y)$ | $\partial_x \psi = 0$ | $\psi = 0$ | $\partial_x \psi = 0$ | $\psi = 0$ |
| $(\mathbf{w}_b,1)$ , $(0,y)$ | $\psi = 0$ | $\partial_x \psi = 0$ | $\psi = 0$ | $\partial_x \psi = 0$ |
| $(\mathbf{w}_b,2)$ , $(0,y)$ | $\partial_x \psi = 0$ | $\psi = 0$ | $\partial_x \psi = 0$ | $\psi = 0$ |
| $(\mathbf{w}_b,3)$ , $(0,y)$ | $\psi = 0$ | $\partial_x \psi = 0$ | $\psi = 0$ | $\partial_x \psi = 0$ |
| $(\mathbf{w}_c,0)$ , $(0,y)$ | $\partial_x \psi = 0$ | $\partial_x \psi = 0$ | $\partial_x \psi = 0$ | $\partial_x \psi = 0$ |
| $(\mathbf{w}_c,1)$ , $(0,y)$ | $\psi = 0$ | $\psi = 0$ | $\psi = 0$ | $\psi = 0$ |
| $(\mathbf{w}_c,2)$ , $(0,y)$ | $\partial_x \psi = 0$ | $\partial_x \psi = 0$ | $\partial_x \psi = 0$ | $\partial_x \psi = 0$ |
| $(\mathbf{w}_c,3)$ , $(0,y)$ | $\psi = 0$ | $\psi = 0$ | $\psi = 0$ | $\psi = 0$ |
| $(\mathbf{w}_d,0)$ , $(0,y)$ | $\partial_x \psi = 0$ | $\psi = 0$ | $\partial_x \psi = 0$ | $\psi = 0$ |
| $(\mathbf{w}_d,1)$ , $(0,y)$ | $\psi = 0$ | $\partial_x \psi = 0$ | $\psi = 0$ | $\partial_x \psi = 0$ |
| $(\mathbf{w}_d,2)$ , $(0,y)$ | $\partial_x \psi = 0$ | $\psi = 0$ | $\partial_x \psi = 0$ | $\partial_x \psi = 0$ |
| $(\mathbf{w}_d,3)$ , $(0,y)$ | $\psi = 0$ | $\partial_x \psi = 0$ | $\psi = 0$ | $\partial_x \psi = 0$ |

In terms of these functions, the normal derivatives at the boundaries become:

$$\mathbf{n}_1 \cdot \nabla \psi^{(\mathbf{w},l)} = \frac{e^{-i\frac{\pi}{2}}}{\sqrt{3}} \left( B_1^{(\mathbf{w},l+1)} - B_2^{(\mathbf{w},l-1)} \right),$$

$$\mathbf{n}_2 \cdot \nabla \psi^{(\mathbf{w},l)} = \frac{1}{\sqrt{3}} \left( e^{i\pi/6} B_1^{(\mathbf{w},l+1)} + e^{-i\pi/6} B_2^{(\mathbf{w},l-1)} \right),$$

$$\mathbf{n}_3 \cdot \nabla \psi^{(\mathbf{w},l)} = \frac{1}{\sqrt{3}} \left( e^{i5\pi/6} B_1^{(\mathbf{w},l+1)} + e^{-i5\pi/6} B_2^{(\mathbf{w},l-1)} \right).$$

The indices $l+1$ and $l-1$ of the functions $B_1$ and $B_2$ indicate that these transform as representations $D_{l\pm1}$ under the operations of the $C_3$ point group. The constraints of equation (5) can therefore also be applied directly to the functions $B_1$ and $B_2$.

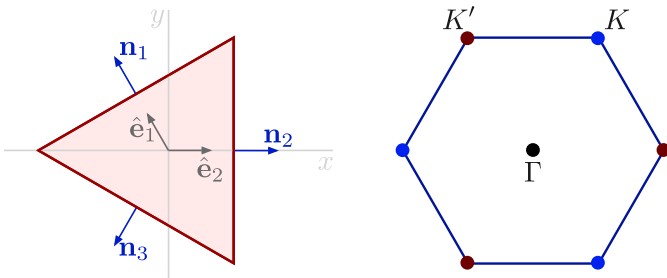

Figure 6: **Left:** a triangular crystal with three-fold rotational symmetry. The unit vectors $\hat{\mathbf{e}}_1$ and $\hat{\mathbf{e}}_2$ are indicated, as well as the normal vectors to the boundaries. **Right:** the corresponding Brillouin zone, with high-symmetry points indicated.

The three-fold rotations leave invariant the Brillouin zone points $\mathbf{\Gamma}$, $\mathbf{K}$, and $\mathbf{K}'$. Symmetry-imposed zeroes of the Bloch wave function or its directional derivatives may occur at these points, for real-space locations coinciding with any of the three Wyckoff positions. They are listed in table 10.

## B  The logarithmic derivative

### B.1  Symmetry-appropriate derivatives

Whenever the point group symmetry of a crystal allows for the Wannier functions to be real-valued, the symmetry transformations may not only impose zeroes on the Bloch wave function and its derivatives, but also guarantee the presence of boundary modes [42]. This is most conveniently seen by considering the logarithmic derivative, defined as:

$$\rho^{(\mathbf{w},l)}(\mathbf{k},\mathbf{r}) = \frac{\mathbf{n}\cdot\nabla\psi^{(\mathbf{w},l)}(\mathbf{k},\mathbf{r})}{\psi^{(\mathbf{w},l)}(\mathbf{k},\mathbf{r})} \,. \tag{7}$$

To predict whether or not boundary modes are present at any given edge of the crystal, we need to consider the transformation properties of the logarithmic derivative under point group operations. We begin by representing the directional derivatives of the Bloch wave function as:

$$B^{(\mathbf{w},\tilde{l})}(\mathbf{k};x,y) = \left(\eta_1\frac{\partial}{\partial x} + \eta_2\frac{\partial}{\partial y}\right)\psi^{(\mathbf{w},l)}(\mathbf{k};x,y)\,. \tag{8}$$

Here, the coefficients $\eta_1$ and $\eta_2$ can be used for example to specify the orientation of the crystal edge.

Table 4: Zeroes imposed by $p_{mm}$-symmetry on either the Bloch wave function or its normal derivative along $\hat{\mathbf{e}}_x$. The real-space positions are fixed at $\mathbf{r} = (a/2, y)$, for arbitrary values of the $y$ coordinate.

| $(w,l)\,,\,\mathbf{r}$ | $\mathbf{k} = \mathbf{\Gamma}$ | $\mathbf{X}$ | $\mathbf{Y}$ | $\mathbf{M}$ |
|---|---|---|---|---|
| $(\mathbf{w}_a,0)\,,\,(\frac{a}{2},y)$ | $\partial_x\psi = 0$ | $\psi = 0$ | $\partial_x\psi = 0$ | $\psi = 0$ |
| $(\mathbf{w}_a,1)\,,\,(\frac{a}{2},y)$ | $\psi = 0$ | $\partial_x\psi = 0$ | $\psi = 0$ | $\partial_x\psi = 0$ |
| $(\mathbf{w}_a,2)\,,\,(\frac{a}{2},y)$ | $\partial_x\psi = 0$ | $\psi = 0$ | $\partial_x\psi = 0$ | $\psi = 0$ |
| $(\mathbf{w}_a,3)\,,\,(\frac{a}{2},y)$ | $\psi = 0$ | $\partial_x\psi = 0$ | $\psi = 0$ | $\partial_x\psi = 0$ |
| $(\mathbf{w}_b,0)\,,\,(\frac{a}{2},y)$ | $\partial_x\psi = 0$ | $\partial_x\psi = 0$ | $\partial_x\psi = 0$ | $\partial_x\psi = 0$ |
| $(\mathbf{w}_b,1)\,,\,(\frac{a}{2},y)$ | $\psi = 0$ | $\psi = 0$ | $\psi = 0$ | $\psi = 0$ |
| $(\mathbf{w}_b,2)\,,\,(\frac{a}{2},y)$ | $\partial_x\psi = 0$ | $\partial_x\psi = 0$ | $\partial_x\psi = 0$ | $\partial_x\psi = 0$ |
| $(\mathbf{w}_b,3)\,,\,(\frac{a}{2},y)$ | $\psi = 0$ | $\psi = 0$ | $\psi = 0$ | $\psi = 0$ |
| $(\mathbf{w}_c,0)\,,\,(\frac{a}{2},y)$ | $\partial_x\psi = 0$ | $\psi = 0$ | $\partial_x\psi = 0$ | $\psi = 0$ |
| $(\mathbf{w}_c,1)\,,\,(\frac{a}{2},y)$ | $\psi = 0$ | $\partial_x\psi = 0$ | $\psi = 0$ | $\partial_x\psi = 0$ |
| $(\mathbf{w}_c,2)\,,\,(\frac{a}{2},y)$ | $\partial_x\psi = 0$ | $\psi = 0$ | $\partial_x\psi = 0$ | $\psi = 0$ |
| $(\mathbf{w}_c,3)\,,\,(\frac{a}{2},y)$ | $\psi = 0$ | $\partial_x\psi = 0$ | $\psi = 0$ | $\partial_x\psi = 0$ |
| $(\mathbf{w}_d,0)\,,\,(\frac{a}{2},y)$ | $\partial_x\psi = 0$ | $\partial_x\psi = 0$ | $\partial_x\psi = 0$ | $\partial_x\psi = 0$ |
| $(\mathbf{w}_d,1)\,,\,(\frac{a}{2},y)$ | $\psi = 0$ | $\psi = 0$ | $\psi = 0$ | $\psi = 0$ |
| $(\mathbf{w}_d,2)\,,\,(\frac{a}{2},y)$ | $\partial_x\psi = 0$ | $\partial_x\psi = 0$ | $\partial_x\psi = 0$ | $\partial_x\psi = 0$ |
| $(\mathbf{w}_d,3)\,,\,(\frac{a}{2},y)$ | $\psi = 0$ | $\psi = 0$ | $\psi = 0$ | $\psi = 0$ |

Table 5: Zeroes imposed by $p_{mm}$-symmetry on either the Bloch wave function or its normal derivative along $\hat{\mathbf{e}}_y$. The real-space positions are fixed at $\mathbf{r} = (x, 0)$, for arbitrary values of the $x$ coordinate.

| $(w, l)$ , $\mathbf{r}$ | $\mathbf{k} = \Gamma$ | X | Y | M |
|---|---|---|---|---|
| $(\mathbf{w}_a, 0)$ , $(x, 0)$ | $\partial_y \psi = 0$ | $\partial_y \psi = 0$ | $\partial_y \psi = 0$ | $\partial_y \psi = 0$ |
| $(\mathbf{w}_a, 1)$ , $(x, 0)$ | $\psi = 0$ | $\psi = 0$ | $\psi = 0$ | $\psi = 0$ |
| $(\mathbf{w}_a, 2)$ , $(x, 0)$ | $\psi = 0$ | $\psi = 0$ | $\psi = 0$ | $\psi = 0$ |
| $(\mathbf{w}_a, 3)$ , $(x, 0)$ | $\partial_y \psi = 0$ | $\partial_y \psi = 0$ | $\partial_y \psi = 0$ | $\partial_y \psi = 0$ |
| $(\mathbf{w}_b, 0)$ , $(x, 0)$ | $\partial_y \psi = 0$ | $\partial_y \psi = 0$ | $\partial_y \psi = 0$ | $\partial_y \psi = 0$ |
| $(\mathbf{w}_b, 1)$ , $(x, 0)$ | $\psi = 0$ | $\psi = 0$ | $\psi = 0$ | $\psi = 0$ |
| $(\mathbf{w}_b, 2)$ , $(x, 0)$ | $\psi = 0$ | $\psi = 0$ | $\psi = 0$ | $\psi = 0$ |
| $(\mathbf{w}_b, 3)$ , $(x, 0)$ | $\partial_y \psi = 0$ | $\partial_y \psi = 0$ | $\partial_y \psi = 0$ | $\partial_y \psi = 0$ |
| $(\mathbf{w}_c, 0)$ , $(x, 0)$ | $\partial_y \psi = 0$ | $\partial_y \psi = 0$ | $\psi = 0$ | $\psi = 0$ |
| $(\mathbf{w}_c, 1)$ , $(x, 0)$ | $\psi = 0$ | $\psi = 0$ | $\partial_y \psi = 0$ | $\partial_y \psi = 0$ |
| $(\mathbf{w}_c, 2)$ , $(x, 0)$ | $\psi = 0$ | $\psi = 0$ | $\partial_y \psi = 0$ | $\partial_y \psi = 0$ |
| $(\mathbf{w}_c, 3)$ , $(x, 0)$ | $\partial_y \psi = 0$ | $\partial_y \psi = 0$ | $\psi = 0$ | $\psi = 0$ |
| $(\mathbf{w}_d, 0)$ , $(x, 0)$ | $\partial_y \psi = 0$ | $\partial_y \psi = 0$ | $\psi = 0$ | $\psi = 0$ |
| $(\mathbf{w}_d, 1)$ , $(x, 0)$ | $\psi = 0$ | $\psi = 0$ | $\partial_y \psi = 0$ | $\partial_y \psi = 0$ |
| $(\mathbf{w}_d, 2)$ , $(x, 0)$ | $\psi = 0$ | $\psi = 0$ | $\partial_y \psi = 0$ | $\partial_y \psi = 0$ |
| $(\mathbf{w}_d, 3)$ , $(x, 0)$ | $\partial_y \psi = 0$ | $\partial_y \psi = 0$ | $\psi = 0$ | $\psi = 0$ |

The functions $B^{(\mathbf{w}, \tilde{l})(\mathbf{k}; x, y)}$ are called symmetry appropriate functions if, just like the Bloch states, they transform as an irreducible representation of the point group keeping the point $(x, y)$ fixed. The index $\tilde{l}$ then indicates that the function $B^{(\mathbf{w}, \tilde{l})}$ transforms as the representation $D_{\tilde{l}}$. Analogous to equation (5), this implies for a symmetry operation $\alpha$ centered at Wyckoff position $\mathbf{w}$ that:

$$\alpha B^{(\mathbf{w}, \tilde{l})}(\mathbf{k}, \mathbf{r}) = \exp(i\mathbf{k} \cdot \mathbf{R}_\mathbf{w}^\alpha) D^{(\tilde{l})}(\alpha) B^{(\mathbf{w}, \tilde{l})}(\mathbf{k}, \mathbf{r}).$$

For the space groups $p_{mm}$ and $p_2$, the characters $D^{(\tilde{l})}(\alpha_\mathbf{w})$ and phase factors $\mathbf{R}_\mathbf{w}^\alpha$ are listed in tables 1, 2, and 7. The constraint that the functions $B^{(\mathbf{w}, \tilde{l})}$ have to both be symmetry-appropriate functions and be related to $\psi^{(\mathbf{w}, l)}$ according to equation (8), yield a one-to-one relation between the indices $\tilde{l}$ of the symmetry-appropriate derivatives, and the indices $l$ of the Bloch wave functions.

For example, if the normal vector to the boundary of a $p_{mm}$-symmetric crystal is along $\mathbf{e}_x$, we can interpret $B^{(\mathbf{w}, \tilde{l})}$ as the normal derivative of the Bloch wave function by taking $\eta_2 = 0$ in equation (8). Considering the mirror operation $\alpha = M_x$ then implies that in order for $B^{(\mathbf{w}, \tilde{l})}$ to be a symmetry-appropriate function, we must have $D^{(\tilde{l})}(M_x) = -D^{(l)}(M_x)$. That is, if $\psi^{(\mathbf{w}, l)}$ is even under the operation $(x, y) \to (-x, y)$, then $B^{(\mathbf{w}, \tilde{l})}$ is odd, and vice versa. In the same way, we can also establish $D^{(\tilde{l})}(M_y) = D^{(l)}(M_y)$ and $D^{(\tilde{l})}(C_2) = -D^{(l)}(C_2)$, and thus we find for this particular termination of a $p_{mm}$ symmetric crystal that $l = 0, 1, 2, 3$ implies $\tilde{l} = 3, 2, 1, 0$.

In $p_2$-symmetric crystals, the connection between Bloch wave functions and their symmetry-appropriate derivatives is even more straightforward. The two-fold rotation affects both the $x$ and $y$ coordinates, so that $D^{(\tilde{l})}(C_2) = -D^{(l)}(C_2)$ and hence $l = 0, 1 \Leftrightarrow \tilde{l} = 1, 0$ for any orientation of the boundary.

## B.2 Gauge transformations and lattice translations

Notice that the logarithmic derivative is invariant under both gauge transformations and translations by any lattice vector. The gauge transformations arise from the fact that Bloch functions for simple bands are defined up to a $\mathbf{k}$ dependent phase factor, and may be written as $\psi^{(\mathbf{w},l)}(\mathbf{k},\mathbf{r}) \to e^{i\phi(\mathbf{k})}\psi^{(\mathbf{w},l)}(\mathbf{k},\mathbf{r})$ [28,53]. Translations by a lattice vector $\mathbf{R}$ similarly affect Bloch functions for simple bands as $\psi^{(\mathbf{w},l)}(\mathbf{k},\mathbf{r}) \to e^{i\mathbf{k}\cdot\mathbf{R}}\psi^{(\mathbf{w},l)}(\mathbf{k},\mathbf{r})$. Since neither of the phase factors used in these transformations depends on the spatial coordinate $\mathbf{r}$, they can be taken outside the gradient in equation (7), and leave the logarithmic derivative invariant.

## B.3 Real-valued Wannier functions

For real-valued Wannier functions, the logarithmic derivative behaves under complex conjugation as [42]:
$$\rho^*(\mathbf{k},\mathbf{r}) = \rho(-\mathbf{k}_R + i\mathbf{k}_I,\mathbf{r}), \quad \text{with} \quad \mathbf{k} = \mathbf{k}_R + i\mathbf{k}_I . \tag{9}$$

Solutions of Schrödinger's equation with complex $\mathbf{k}$ here represent boundary modes that decay exponentially in real space.

Both $p_{mm}$ and $p_2$ allow real-valued Wannier functions. They share the same four high-symmetry points in reciprocal space, all of which have the property that $\mathbf{k}_R = -\mathbf{k}_R$ up to translations by a reciprocal lattice vector. From equation (9) it is then clear that the logarithmic derivative at these points is a real-valued function. This includes in particular any boundary modes that arise at high symmetry values of $\mathbf{k}_R$.

Combining the transformation of the logarithmic derivative under complex conjugation with its transformation properties under symmetry group operations restricts the logarithmic

Table 6: Zeroes imposed by $p_{mm}$-symmetry on either the Bloch wave function or its normal derivative along $\hat{\mathbf{e}}_y$. The real-space positions are fixed at $\mathbf{r} = (x, a/2)$, for arbitrary values of the $x$ coordinate.

| $(w,l)$, $\mathbf{r}$ | $\mathbf{k} = \Gamma$ | X | Y | M |
|---|---|---|---|---|
| $(\mathbf{w}_a,0)$, $(x,\frac{a}{2})$ | $\partial_y\psi = 0$ | $\partial_y\psi = 0$ | $\psi = 0$ | $\psi = 0$ |
| $(\mathbf{w}_a,1)$, $(x,\frac{a}{2})$ | $\psi = 0$ | $\psi = 0$ | $\partial_y\psi = 0$ | $\partial_y\psi = 0$ |
| $(\mathbf{w}_a,2)$, $(x,\frac{a}{2})$ | $\psi = 0$ | $\psi = 0$ | $\partial_y\psi = 0$ | $\partial_y\psi = 0$ |
| $(\mathbf{w}_a,3)$, $(x,\frac{a}{2})$ | $\partial_y\psi = 0$ | $\partial_y\psi = 0$ | $\psi = 0$ | $\psi = 0$ |
| $(\mathbf{w}_b,0)$, $(x,\frac{a}{2})$ | $\partial_y\psi = 0$ | $\partial_y\psi = 0$ | $\psi = 0$ | $\psi = 0$ |
| $(\mathbf{w}_b,1)$, $(x,\frac{a}{2})$ | $\psi = 0$ | $\psi = 0$ | $\partial_y\psi = 0$ | $\partial_y\psi = 0$ |
| $(\mathbf{w}_b,2)$, $(x,\frac{a}{2})$ | $\psi = 0$ | $\psi = 0$ | $\partial_y\psi = 0$ | $\partial_y\psi = 0$ |
| $(\mathbf{w}_b,3)$, $(x,\frac{a}{2})$ | $\partial_y\psi = 0$ | $\partial_y\psi = 0$ | $\psi = 0$ | $\psi = 0$ |
| $(\mathbf{w}_c,0)$, $(x,\frac{a}{2})$ | $\partial_y\psi = 0$ | $\partial_y\psi = 0$ | $\partial_y\psi = 0$ | $\partial_y\psi = 0$ |
| $(\mathbf{w}_c,1)$, $(x,\frac{a}{2})$ | $\psi = 0$ | $\psi = 0$ | $\psi = 0$ | $\psi = 0$ |
| $(\mathbf{w}_c,2)$, $(x,\frac{a}{2})$ | $\psi = 0$ | $\psi = 0$ | $\psi = 0$ | $\psi = 0$ |
| $(\mathbf{w}_c,3)$, $(x,\frac{a}{2})$ | $\partial_y\psi = 0$ | $\partial_y\psi = 0$ | $\partial_y\psi = 0$ | $\partial_y\psi = 0$ |
| $(\mathbf{w}_d,0)$, $(x,\frac{a}{2})$ | $\partial_y\psi = 0$ | $\partial_y\psi = 0$ | $\partial_y\psi = 0$ | $\partial_y\psi = 0$ |
| $(\mathbf{w}_d,1)$, $(x,\frac{a}{2})$ | $\psi = 0$ | $\psi = 0$ | $\psi = 0$ | $\psi = 0$ |
| $(\mathbf{w}_d,2)$, $(x,\frac{a}{2})$ | $\psi = 0$ | $\psi = 0$ | $\psi = 0$ | $\psi = 0$ |
| $(\mathbf{w}_d,3)$, $(x,\frac{a}{2})$ | $\partial_y\psi = 0$ | $\partial_y\psi = 0$ | $\partial_y\psi = 0$ | $\partial_y\psi = 0$ |

Table 7: The character table and phase factors associated with the point group $C_2$, relevant at Wyckoff positions in a $p_2$-symmetric atomic insulator.

|       | $E$ | $C_2$ | $\mathbf{w}$ | $\mathbf{R}_{\mathbf{w}}^{C_2}$ |
|-------|-----|-------|--------------|----------------------------------|
| $D_0$ | 1   | 1     | $\mathbf{w}_a = (0,0)$ | $\mathbf{0}$ |
| $D_1$ | 1   | -1    | $\mathbf{w}_b = (\frac{a}{2},0)$ | $a\mathbf{e}_x$ |
|       |     |       | $\mathbf{w}_c = (0,\frac{a}{2})$ | $a\mathbf{e}_y$ |
|       |     |       | $\mathbf{w}_d = (\frac{a}{2},\frac{a}{2})$ | $a\mathbf{e}_x + a\mathbf{e}_y$ |

derivative even further. Since both the Bloch wave functions $\psi^{(\mathbf{w},l)}$ and their derivatives $B^{(\mathbf{w},\tilde{l})}$ are constructed from Wannier functions centered at the Wyckoff position $\mathbf{w}$, they share the same associated phase factors $e^{i\mathbf{k}\cdot\mathbf{R}_{\mathbf{w}}^{\alpha}}$ for point group operations centered at $\mathbf{w}$ [33]. Applying equation (5) to the logarithmic derivative then yields:

$$\rho(\alpha^{-1}\mathbf{k}, \alpha^{-1}\mathbf{r}) = \frac{D^{(\tilde{l})}(\alpha)}{D^{(l)}(\alpha)}\rho(\mathbf{k},\mathbf{r}). \tag{10}$$

For the special case in which the momentum $\mathbf{k}$ is purely real and the real-space position $\mathbf{r}$ is a Wyckoff position, this relation reduces to:

$$\rho(\alpha^{-1}\mathbf{k}_R, \mathbf{w}) = \frac{D^{(\tilde{l})}(\alpha)}{D^{(l)}(\alpha)}\rho(\mathbf{k}_R,\mathbf{w}). \tag{11}$$

Considering for example a $p_2$-symmetric crystal, and taking $\alpha$ to be the two-fold rotation yields $\alpha^{-1}\mathbf{k}_R = -\mathbf{k}_R$ and $D^{(\tilde{l})}(\alpha)/D^{(l)}(\alpha) = -1$. Combining equations (9) and (11) then leads to the constraint that the logarithmic derivative must be purely imaginary for any real $\mathbf{k}$. The same argument applies to any crystal with inversion symmetry, while considering only a single mirror operation (say $M_x$) suffices to render the logarithmic derivative purely imaginary only for real momenta along lines in the Brillouin zone ($k_y = 0$ and $k_y = \pi$) and for specific orientations of the crystal boundary (parallel to $\hat{\mathbf{e}}_y$).

As explained in the main text, the constraints discussed above for crystals whose space group includes the inversion operation and allows for real-valued Wannier functions [54], suffice to determine whether or not the logarithmic derivative changes sign from one high-symmetry momentum value to another. This in turn determines whether or not any boundary states arise at the high-symmetry momenta.

### B.4 The sign of the logarithmic derivative

To predict the presence of interface states at junction interfaces, we first establish how the signs of the logarithmic derivative for in-gap states depends on its zeroes and divergences at high-symmetry points. For concreteness, we consider momentum values going from $\mathbf{\Gamma} = (0,0)$ to $\mathbf{X} = (\pi/a, 0)$, but the same reasoning can be applied to any other pair of high symmetry points. Whether $\rho = 0$ or $\rho = \infty$ for $\mathbf{k}$ and $\mathbf{r}$ fixed at high symmetry positions (in the Brillouin zone and real space respectively), can be deduced from tables 3 to 8.

First, consider the case of $\rho(\mathbf{\Gamma}, \mathbf{r}) = 0$, with $\mathbf{r}$ a high-symmetry real-space position at the boundary of the crystal. For the logarithmic derivative to vanish, we must have that $B^{(\mathbf{w},\tilde{l})}(\mathbf{k},\mathbf{r}) = B^{(\mathbf{w},\tilde{l})}([k_x, k_y],\mathbf{r})$ vanishes as well. We can then use the analytic nature of $B^{(\mathbf{w},\tilde{l})}$

to write it as a power series for momenta close to $\Gamma$ and approximate it as:

$$B^{(\mathbf{w},\tilde{l})}([k_x,0],\mathbf{r}) = \sum_n c_n k_x^n$$

$$\Rightarrow B^{(\mathbf{w},\tilde{l})}([\pm i\delta,0],\mathbf{r}) \sim \delta, \quad \text{for} \quad \delta << 1. \tag{12}$$

Here, we used the fact that for in-gap states, the logarithmic derivative is real, and we assumed it to be positive in this example, without loss of generality. Using continuity, it immediately follows that $B^{(\mathbf{w},\tilde{l})}([\delta,0],\mathbf{r}) \sim \mp i\delta$.

Given that the logarithmic derivative for real momenta is purely imaginary, $\rho$ remains

Table 8: Zeroes imposed by $p_2$-symmetry on the Bloch function $\psi$ and its normal derivative $\psi'$ in any direction, for all high-symmetry points in the Brillouin zone and real space.

| $(w,l)$ , $\mathbf{r}$ | $\mathbf{k} = \Gamma$ | X | Y | M |
|---|---|---|---|---|
| $(\mathbf{w}_a,0)$ , $(0,0)$ | $\psi'=0$ | $\psi'=0$ | $\psi'=0$ | $\psi'=0$ |
| $(\mathbf{w}_a,1)$ , $(0,0)$ | $\psi=0$ | $\psi=0$ | $\psi=0$ | $\psi=0$ |
| $(\mathbf{w}_b,0)$ , $(0,0)$ | $\psi'=0$ | $\psi=0$ | $\psi'=0$ | $\psi=0$ |
| $(\mathbf{w}_b,1)$ , $(0,0)$ | $\psi=0$ | $\psi'=0$ | $\psi=0$ | $\psi'=0$ |
| $(\mathbf{w}_c,0)$ , $(0,0)$ | $\psi'=0$ | $\psi'=0$ | $\psi=0$ | $\psi=0$ |
| $(\mathbf{w}_c,1)$ , $(0,0)$ | $\psi=0$ | $\psi=0$ | $\psi'=0$ | $\psi'=0$ |
| $(\mathbf{w}_d,0)$ , $(0,0)$ | $\psi'=0$ | $\psi=0$ | $\psi=0$ | $\psi'=0$ |
| $(\mathbf{w}_d,1)$ , $(0,0)$ | $\psi=0$ | $\psi'=0$ | $\psi'=0$ | $\psi=0$ |
| $(\mathbf{w}_a,0)$ , $(\frac{a}{2},0)$ | $\psi'=0$ | $\psi=0$ | $\psi'=0$ | $\psi=0$ |
| $(\mathbf{w}_a,1)$ , $(\frac{a}{2},0)$ | $\psi=0$ | $\psi'=0$ | $\psi=0$ | $\psi'=0$ |
| $(\mathbf{w}_b,0)$ , $(\frac{a}{2},0)$ | $\psi'=0$ | $\psi'=0$ | $\psi'=0$ | $\psi'=0$ |
| $(\mathbf{w}_b,1)$ , $(\frac{a}{2},0)$ | $\psi=0$ | $\psi=0$ | $\psi=0$ | $\psi=0$ |
| $(\mathbf{w}_c,0)$ , $(\frac{a}{2},0)$ | $\psi'=0$ | $\psi=0$ | $\psi=0$ | $\psi'=0$ |
| $(\mathbf{w}_c,1)$ , $(\frac{a}{2},0)$ | $\psi=0$ | $\psi'=0$ | $\psi'=0$ | $\psi=0$ |
| $(\mathbf{w}_d,0)$ , $(\frac{a}{2},0)$ | $\psi'=0$ | $\psi'=0$ | $\psi=0$ | $\psi=0$ |
| $(\mathbf{w}_d,1)$ , $(\frac{a}{2},0)$ | $\psi=0$ | $\psi=0$ | $\psi'=0$ | $\psi'=0$ |
| $(\mathbf{w}_a,0)$ , $(0,\frac{a}{2})$ | $\psi'=0$ | $\psi'=0$ | $\psi=0$ | $\psi=0$ |
| $(\mathbf{w}_a,1)$ , $(0,\frac{a}{2})$ | $\psi=0$ | $\psi=0$ | $\psi'=0$ | $\psi'=0$ |
| $(\mathbf{w}_b,0)$ , $(0,\frac{a}{2})$ | $\psi'=0$ | $\psi=0$ | $\psi=0$ | $\psi'=0$ |
| $(\mathbf{w}_b,1)$ , $(0,\frac{a}{2})$ | $\psi=0$ | $\psi'=0$ | $\psi'=0$ | $\psi=0$ |
| $(\mathbf{w}_c,0)$ , $(0,\frac{a}{2})$ | $\psi'=0$ | $\psi'=0$ | $\psi'=0$ | $\psi'=0$ |
| $(\mathbf{w}_c,1)$ , $(0,\frac{a}{2})$ | $\psi=0$ | $\psi=0$ | $\psi=0$ | $\psi=0$ |
| $(\mathbf{w}_d,0)$ , $(0,\frac{a}{2})$ | $\psi'=0$ | $\psi=0$ | $\psi'=0$ | $\psi=0$ |
| $(\mathbf{w}_d,1)$ , $(0,\frac{a}{2})$ | $\psi=0$ | $\psi'=0$ | $\psi=0$ | $\psi'=0$ |
| $(\mathbf{w}_a,0)$ , $(\frac{a}{2},\frac{a}{2})$ | $\psi'=0$ | $\psi=0$ | $\psi=0$ | $\psi'=0$ |
| $(\mathbf{w}_a,1)$ , $(\frac{a}{2},\frac{a}{2})$ | $\psi=0$ | $\psi'=0$ | $\psi'=0$ | $\psi=0$ |
| $(\mathbf{w}_b,0)$ , $(\frac{a}{2},\frac{a}{2})$ | $\psi'=0$ | $\psi'=0$ | $\psi=0$ | $\psi=0$ |
| $(\mathbf{w}_b,1)$ , $(\frac{a}{2},\frac{a}{2})$ | $\psi=0$ | $\psi=0$ | $\psi'=0$ | $\psi'=0$ |
| $(\mathbf{w}_c,0)$ , $(\frac{a}{2},\frac{a}{2})$ | $\psi'=0$ | $\psi=0$ | $\psi'=0$ | $\psi=0$ |
| $(\mathbf{w}_c,1)$ , $(\frac{a}{2},\frac{a}{2})$ | $\psi=0$ | $\psi'=0$ | $\psi=0$ | $\psi'=0$ |
| $(\mathbf{w}_d,0)$ , $(\frac{a}{2},\frac{a}{2})$ | $\psi'=0$ | $\psi'=0$ | $\psi'=0$ | $\psi'=0$ |
| $(\mathbf{w}_d,1)$ , $(\frac{a}{2},\frac{a}{2})$ | $\psi=0$ | $\psi=0$ | $\psi=0$ | $\psi=0$ |

Table 9: The character table and phase factors associated with the point group $C_3$, relevant at Wyckoff positions in a $p_3$-symmetric atomic insulator.

| | $E$ | $C_3$ | $C_3^2$ |
|---|---|---|---|
| $D_0$ | 1 | 1 | 1 |
| $D_1$ | 1 | $e^{i\frac{2\pi}{3}}$ | $e^{-i\frac{2\pi}{3}}$ |
| $D_2$ | 1 | $e^{-i\frac{2\pi}{3}}$ | $e^{i\frac{2\pi}{3}}$ |

| $\mathbf{w}$ | $\mathbf{R_w}^{C_3}$ | $\mathbf{R_w}^{C_3^2}$ |
|---|---|---|
| $\mathbf{w}_a = (0,0)$ | $\mathbf{0}$ | $\mathbf{0}$ |
| $\mathbf{w}_b = (\frac{2a}{3}, \frac{a}{3})$ | $a\hat{\mathbf{e}}_1$ | $a(\hat{\mathbf{e}}_1 + \hat{\mathbf{e}}_2)$ |
| $\mathbf{w}_c = (\frac{a}{3}, \frac{2a}{3})$ | $a(\hat{\mathbf{e}}_1 + \hat{\mathbf{e}}_2)$ | $a\hat{\mathbf{e}}_2$ |

Table 10: Zeroes of the Bloch wave function $\psi$ and its directional derivatives $\mathbf{n}_i \cdot \nabla \psi$, where $\mathbf{n}_i$ can be any of three vectors normal to the crystal boundaries. The zeroes are listed for all high-symmetry points in the Brillouin zone, and real space positions $\mathbf{r} = (0,0)$, $\mathbf{r} = (2a/3, a/3)$, and $\mathbf{r} = (a/3, 2a/3)$.

| $(w,l)$, $\mathbf{r}$ | $\mathbf{k} = \Gamma$ | $K$ | $K'$ |
|---|---|---|---|
| $(\mathbf{w}_a,0)$ , $(0,0)$ | $\mathbf{n}_i.\nabla\psi = 0$ | $\mathbf{n}_i.\nabla\psi = 0$ | $\mathbf{n}_i.\nabla\psi = 0$ |
| $(\mathbf{w}_a,1)$ , $(0,0)$ | $\psi = 0$ | $\psi = 0$ | $\psi = 0$ |
| $(\mathbf{w}_a,2)$ , $(0,0)$ | $\psi = 0$ | $\psi = 0$ | $\psi = 0$ |
| $(\mathbf{w}_b,0)$ , $(0,0)$ | $\mathbf{n}_i.\nabla\psi = 0$ | $\mathbf{n}_i.\nabla\psi = 0$ | $\mathbf{n}_i.\nabla\psi = 0$ |
| $(\mathbf{w}_b,1)$ , $(0,0)$ | $\psi = 0$ | $\psi = 0$ | $\psi = 0$ |
| $(\mathbf{w}_b,2)$ , $(0,0)$ | $\psi = 0$ | $\psi = 0$ | $\psi = 0$ |
| $(\mathbf{w}_c,0)$ , $(0,0)$ | $\mathbf{n}_i.\nabla\psi = 0$ | $\mathbf{n}_i.\nabla\psi = 0$ | $\mathbf{n}_i.\nabla\psi = 0$ |
| $(\mathbf{w}_c,1)$ , $(0,0)$ | $\psi = 0$ | $\psi = 0$ | $\psi = 0$ |
| $(\mathbf{w}_c,2)$ , $(0,0)$ | $\psi = 0$ | $\psi = 0$ | $\psi = 0$ |
| $(\mathbf{w}_a,0)$ , $(\frac{2a}{3}, \frac{a}{3})$ | $\mathbf{n}_i.\nabla\psi = 0$ | $\psi = 0$ | $\psi = 0$ |
| $(\mathbf{w}_a,1)$ , $(\frac{2a}{3}, \frac{a}{3})$ | $\psi = 0$ | $\psi = 0$ | $\mathbf{n}_i.\nabla\psi = 0$ |
| $(\mathbf{w}_a,2)$ , $(\frac{2a}{3}, \frac{a}{3})$ | $\psi = 0$ | $\mathbf{n}_i.\nabla\psi = 0$ | $\psi = 0$ |
| $(\mathbf{w}_b,0)$ , $(\frac{2a}{3}, \frac{a}{3})$ | $\mathbf{n}_i.\nabla\psi = 0$ | $\mathbf{n}_i.\nabla\psi = 0$ | $\mathbf{n}_i.\nabla\psi = 0$ |
| $(\mathbf{w}_b,1)$ , $(\frac{2a}{3}, \frac{a}{3})$ | $\psi = 0$ | $\psi = 0$ | $\psi = 0$ |
| $(\mathbf{w}_b,2)$ , $(\frac{2a}{3}, \frac{a}{3})$ | $\psi = 0$ | $\psi = 0$ | $\psi = 0$ |
| $(\mathbf{w}_c,0)$ , $(\frac{2a}{3}, \frac{a}{3})$ | $\mathbf{n}_i.\nabla\psi = 0$ | $\psi = 0$ | $\psi = 0$ |
| $(\mathbf{w}_c,1)$ , $(\frac{2a}{3}, \frac{a}{3})$ | $\psi = 0$ | $\mathbf{n}_i.\nabla\psi = 0$ | $\psi = 0$ |
| $(\mathbf{w}_c,2)$ , $(\frac{2a}{3}, \frac{a}{3})$ | $\psi = 0$ | $\psi = 0$ | $\mathbf{n}_i.\nabla\psi = 0$ |
| $(\mathbf{w}_a,0)$ , $(\frac{a}{3}, \frac{2a}{3})$ | $\mathbf{n}_i.\nabla\psi = 0$ | $\psi = 0$ | $\psi = 0$ |
| $(\mathbf{w}_a,1)$ , $(\frac{a}{3}, \frac{2a}{3})$ | $\psi = 0$ | $\mathbf{n}_i.\nabla\psi = 0$ | $\psi = 0$ |
| $(\mathbf{w}_a,2)$ , $(\frac{a}{3}, \frac{2a}{3})$ | $\psi = 0$ | $\psi = 0$ | $\mathbf{n}_i.\nabla\psi = 0$ |
| $(\mathbf{w}_b,0)$ , $(\frac{a}{3}, \frac{2a}{3})$ | $\mathbf{n}_i.\nabla\psi = 0$ | $\psi = 0$ | $\psi = 0$ |
| $(\mathbf{w}_b,1)$ , $(\frac{a}{3}, \frac{2a}{3})$ | $\psi = 0$ | $\psi = 0$ | $\mathbf{n}_i.\nabla\psi = 0$ |
| $(\mathbf{w}_b,2)$ , $(\frac{a}{3}, \frac{2a}{3})$ | $\psi = 0$ | $\mathbf{n}_i.\nabla\psi = 0$ | $\psi = 0$ |
| $(\mathbf{w}_c,0)$ , $(\frac{a}{3}, \frac{2a}{3})$ | $\mathbf{n}_i.\nabla\psi = 0$ | $\mathbf{n}_i.\nabla\psi = 0$ | $\mathbf{n}_i.\nabla\psi = 0$ |
| $(\mathbf{w}_c,1)$ , $(\frac{a}{3}, \frac{2a}{3})$ | $\psi = 0$ | $\psi = 0$ | $\psi = 0$ |
| $(\mathbf{w}_c,2)$ , $(\frac{a}{3}, \frac{2a}{3})$ | $\psi = 0$ | $\psi = 0$ | $\psi = 0$ |

constrained to the imaginary axis as we transverse the band, and can only change sign when either $B^{(\mathbf{w},\tilde{l})}$ or $\psi^{(\mathbf{w},l)}$ goes to zero. This is forced by symmetry to happen again only at $\mathbf{k} = \mathbf{X}$, where we can again introduce a power series expansion:

$$B^{(\mathbf{w},\tilde{l})}([k_x,0],\mathbf{r}) = \sum_n d_n (k_x - \pi/a)^n \,. \tag{13}$$

Considering the case in which the logarithmic derivative vanishes at $\mathbf{X}$, and using the fact that $\rho$ could not change sign while traversing the band from $\mathbf{\Gamma}$ to $\mathbf{X}$, then yields:

$$B^{(\mathbf{w},\tilde{l})}([\pi/a-\delta,0],\mathbf{r}) \sim \mp i\delta \,,$$
$$B^{(\mathbf{w},\tilde{l})}([\pi/a\pm i\delta,0],\mathbf{r}) \sim -\delta \,, \quad \text{for} \quad \delta << 1, \tag{14}$$

where in the final line, we again used continuity of $B^{(\mathbf{w},\tilde{l})}$. The logarithmic derivatives in the gaps bordering $\mathbf{\Gamma}$ and $\mathbf{X}$ thus have opposite sign, and more generally, the sign of $\rho$ for in-gap states changes from one gap to the next if the gaps are connected by a band with vanishing logarithmic derivatives at both high-symmetry momenta bordering the gaps. Note that for a given position of the boundary, tables 3 to 8 can be used to identify the band labels $(\mathbf{w},l)$ that yield zeroes at any pair of high symmetry momenta.

Next, consider the case of the logarithmic derivative diverging at both high symmetry momenta. The same type of series expansions can now be made for the wave function $\psi^{(\mathbf{w},l)}$ rather than the symmetry-appropriate derivative. Close to $\mathbf{\Gamma}$, we can then start from $\psi^{(\mathbf{w},l)}([\pm i\delta,0],\mathbf{r}) \sim \delta \Rightarrow \rho([\delta,0],\mathbf{r}) \sim \pm i/\delta$. Following the same reasoning as before, this in turn implies $\rho([\pi/a-\delta,0],\mathbf{r}) \sim \pm i/\delta$, and hence $\rho([\pi/a\pm\delta,0],\mathbf{r}) \sim -1/\delta$. Thus, the logarithmic derivative has opposite signs in two gaps connected by a band with equal behaviour of the logarithmic derivative at its two endpoints, regardless of whether it goes to zero or diverges.

The final case to consider has a vanishing logarithmic derivative at one high symmetry momentum, and a divergence at the other. For concreteness we take $\mathbf{\Gamma}$ to have $\rho = 0$ while $\rho = \infty$ at $\mathbf{X}$, but the result is general. We then start with $\rho([\pm i\delta,0],\mathbf{r}) \sim \delta$, leading to $\rho([\delta,0],\mathbf{r}) \sim \mp i\delta$. This fixes the sign of the purely imaginary logarithmic derivative as we traverse the band from $\mathbf{\Gamma}$ to $\mathbf{X}$. Close to $\mathbf{X}$, where $\rho$ diverges, we then find $\rho([\pi/a-\delta,0],\mathbf{r}) \sim \mp i/\delta$. Continuity finally allows us to infer that $\rho([\pi/a\pm i\delta,0],\mathbf{r}) \sim 1/\delta$, and hence that the logarithmic derivative for in-gap states close to $\mathbf{X}$ has the same sign as for in-gap states close to $\mathbf{\Gamma}$. More generally, we thus find that there are two possible scenarios: if a gap at one high symmetry momentum is connected to a gap at another high symmetry point by a band whose logarithmic derivative has the same behaviour at both endpoints, the signs of $\rho$ for the in-gap states are opposite, while bands with different behavior of the logarithmic derivative at the high symmetry points yield equal signs for the in-gap states.

## B.5 Independence from boundary shifts in $p_2$ and $p_{mm}$

Whether the signs of logarithmic derivatives match across an interface separating two atomic insulators, depends on the location of the interface as well as on the band structures of the two insulators. Nevertheless, the prediction of whether or not interface states exist in junctions between two atomic insulators with the same space group, is robust to spatial shifts (or distortions) of the interface from one Wyckoff position to any other. To see this, consider one band each in two insulators joined together in a heterojunction architecture (as shown in Fig. 7). The bands in the two insulators in the top panel have different limiting behaviours for their logarithmic derivatives. The material on the left has vanishing $\rho$ at both $\mathbf{\Gamma}$ and $\mathbf{X}$, while the material on the right has one zero and one divergence.

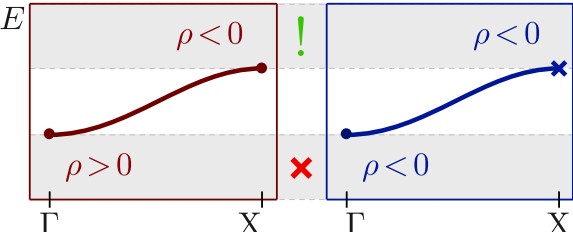

Figure 7: An examples of energy bands of neighbouring crystalline insulators in a heterojunction architecture. The indicated signs of the logarithmic derivatives $\rho$ for in-gap states are uniquely determine the zeroes of the Bloch wave function (solid dots) and its derivative (crosses) at high-symmetry points in the Brillouin zone. A topological obstruction for bulk states to connect across the interface arises when the signs of the logarithmic derivatives within a particular gap agree across the junction, as for the top gap. A localised topological state then emerges at the interface.

The zeroes of the Bloch function $\psi^{(\mathbf{w},l)}$ and its symmetry-appropriate derivative $B^{(\mathbf{w},\tilde{l})}$ are determined by their symmetry transformations, which read:

$$\left(1 - e^{i\mathbf{k}\cdot(\mathbf{R}_{\mathbf{w}}^{\alpha} - \mathbf{R})} D^{(l)}(\alpha)\right) \psi^{(\mathbf{w},l)}(\mathbf{k},\mathbf{r}) = 0,$$
$$\left(1 - e^{i\mathbf{k}\cdot(\mathbf{R}_{\mathbf{w}}^{\alpha} - \mathbf{R})} D^{(\tilde{l})}(\alpha)\right) B^{(\mathbf{w},\tilde{l})}(\mathbf{k},\mathbf{r}) = 0. \tag{15}$$

Here $\mathbf{R}$ is a reciprocal lattice vector, defined through the relation $\alpha^{-1}\mathbf{r} = \mathbf{r} + \mathbf{R}$. In $p2$ and $pmm$-symmetric crystals, the symmetry operations require either the Bloch function or its derivative to go to zero at high symmetry momenta. We must therefore either have $e^{i\mathbf{k}\cdot(\mathbf{R}_{\mathbf{w}}^{\alpha} - \mathbf{R})} D^{(l)}(\alpha) = 1$, or $e^{i\mathbf{k}\cdot(\mathbf{R}_{\mathbf{w}}^{\alpha} - \mathbf{R})} D^{(\tilde{l})}(\alpha) = 1$ at those positions in the Brillouin zone.

If we now change $\mathbf{r}$ from one high symmetry position in real space to another, the only effect this can have in Eq. (15), is to change the value of $\mathbf{R}$ in the phase factor. This change upon moving the location of the interface from $\mathbf{r}$ to $\mathbf{r}' = \mathbf{r} + \delta\mathbf{r}$ is determined by action of the symmetry operations: $\mathbf{R}' = \alpha^{-1}\mathbf{r}' - \mathbf{r}' = \mathbf{R} + \alpha^{-1}\delta\mathbf{r} - \delta\mathbf{r}$. The two crystals on opposite sides of the junction thus both acquire an additional term $\mathbf{k}\cdot(\alpha^{-1}\delta\mathbf{r} - \delta\mathbf{r})$ in their phase factors. For a given high symmetry momentum value, if for example the condition $e^{i\mathbf{k}\cdot(\mathbf{R}_{\mathbf{w}}^{\alpha} - \mathbf{R})} D^{(l)}(\alpha) = 1$ was satisfied with the interface at $\mathbf{r}$, the additional phase factor associated with moving the interface to $\mathbf{r}'$ can either cause the condition to remain satisfied, or to be violated. Since there must always be a zero of either the Bloch function or its derivative, the latter case would cause the condition $e^{i\mathbf{k}\cdot(\mathbf{R}_{\mathbf{w}}^{\alpha} - \mathbf{R}')} D^{(\tilde{l})}(\alpha) = 1$ on the representation of the derivative to become satisfied. But since the crystal on the other side of the junction incurs the same additions to its phase factors, it must undergo the same pattern of either keeping its zero at $\mathbf{k}$ fixed or shifting it between the Bloch function and its derivative.

This argument holds for all high symmetry momenta, and thus guarantees that at any given point in the Brillouin zone either the zeroes and divergences on both sides of the junction are unaltered by a shift of the interface, or they change simultaneously. Either way, whether or not the signs of the in-gap logarithmic derivatives change from the gap below the band to the gap above it, is affected in the same way on either side of the junction. A gap with matching signs for an interface at $\mathbf{r}$ will therefore also have matching signs with the interface at $\mathbf{r}'$, and similarly for gaps in which the signs do not match. Changing the location of the interface, from one high symmetry location in real space to another therefore does not affect the existence of interface states for junctions in which the crystals are either both $p2$ or both $pmm$-symmetric.

Table 11: Correspondence between real-space symmetry labels and momentum-space irreducible representations in a $p2$-symmetric atomic insulator.

| $(w,l)$ | $\Gamma$ | $X$ | $Y$ | $M$ |
|---|---|---|---|---|
| $(\mathbf{w}_a,0)$ | $\Gamma_0$ | $X_0$ | $Y_0$ | $M_0$ |
| $(\mathbf{w}_a,1)$ | $\Gamma_1$ | $X_1$ | $Y_1$ | $M_1$ |
| $(\mathbf{w}_b,0)$ | $\Gamma_0$ | $X_1$ | $Y_0$ | $M_1$ |
| $(\mathbf{w}_b,1)$ | $\Gamma_1$ | $X_0$ | $Y_1$ | $M_0$ |
| $(\mathbf{w}_c,0)$ | $\Gamma_0$ | $X_0$ | $Y_1$ | $M_1$ |
| $(\mathbf{w}_c,1)$ | $\Gamma_1$ | $X_1$ | $Y_0$ | $M_0$ |
| $(\mathbf{w}_d,0)$ | $\Gamma_0$ | $X_1$ | $Y_1$ | $M_0$ |
| $(\mathbf{w}_d,1)$ | $\Gamma_1$ | $X_0$ | $Y_0$ | $M_1$ |

## C  Momentum-space irreducible representations

For atomic insulators it is always possible to induce the momentum space representations at high-symmetry points from the real-space band labels [34,47]. Notice however that the former are more general, in the sense that they describe strong topological insulators with non-zero Chern numbers as well as atomic insulators, whereas the real-space band symmetry labels can be applied only to atomic insulators [20, 32, 41]. For crystals with $p2$ or $pmm$ symmetry, we can straightforwardly construct the explicit map from real space to momentum space labels.

### C.1  $p_2$-symmetric atomic insulators

For $p_2$-symmetric atomic insulators, the irreducible representations of Bloch states at high symmetry points in the Brillouin zone can be deduced directly from the transformation properties associated with the real-space symmetry labels. As an example, consider the wave function $\psi^{(\mathbf{w}_a,l)}(\mathbf{k},\mathbf{r})$. The only non-trivial symmetry operation that we can consider for $p2$-symmetric crystals is the two-fold rotation $C_2$. From Eq. (5), we can see that at for example $\mathbf{k} = \Gamma$, this symmetry operation acts on the wave function as:

$$
\begin{aligned}
C_2\psi^{(\mathbf{w}_a,l)}(\Gamma,\mathbf{r}) &= D_l(C_2)\psi^{(\mathbf{w}_a,l)}(\Gamma,\mathbf{r}), \\
&= (-1)^l \psi^{(\mathbf{w}_a,l)}(\Gamma,\mathbf{r}).
\end{aligned}
\tag{16}
$$

At the momentum point $\Gamma$, the wave function is thus even under inversion for the real-space symmetry label with $l = 0$, and odd for $l = 1$. Consequently, a band with real-space symmetry label $(\mathbf{w}_a, 0)$ gives rise to the irreducible representation $\Gamma_0$ (with character $+1$) at the momentum point $\Gamma$, while $(\mathbf{w}_a, 1)$ generates to the irreducible representation $\Gamma_1$ (with character $-1$). Proceeding in this way for the other high symmetry points in the Brillouin zone, we can compute what is the string of irreducible representations in momentum space produced by all the different $(\mathbf{w},l)$ elementary band representations. The results are illustrated in Table 11, which is consistent with [32].

Notice that Table 11 contains only half of all possible $2^4 = 16$ combinations of even and odd irreducible representations at the four high symmetry points in the Brillouin zone. The combinations in the table are the only ones consistent with Wannier representations indexed by real space symmetry labels $(\mathbf{w},l)$, and thus the only ones relevant for atomic insulators. The missing combinations of momentum-space irreducible representations represent bands with a non-zero Chern number, which cannot be written in terms of globally defined localised Wannier functions. Here, we focus on the atomic insulators, with zero Chern number.

Table 12: Correspondence between real-space symmetry labels and momentum-space irreducible representations in a *pmm*-symmetric atomic insulator. The entry $\{l_1, l_2, l_3, l_4\}$ lists the induced irreducible representations of the high symmetry lines.

| $(w,l)$ | $\{l_1, l_2, l_3, l_4\}$ | $\mathbf{\Gamma}$ | $\mathbf{X}$ | $\mathbf{Y}$ | $\mathbf{M}$ |
|---|---|---|---|---|---|
| $(\mathbf{w}_a, 0)$ | $\{+,+,+,+\}$ | $\Gamma_0$ | $X_0$ | $Y_0$ | $M_0$ |
| $(\mathbf{w}_a, 1)$ | $\{-,-,-,-\}$ | $\Gamma_1$ | $X_1$ | $Y_1$ | $M_1$ |
| $(\mathbf{w}_a, 2)$ | $\{-,+,-,+\}$ | $\Gamma_2$ | $X_2$ | $Y_2$ | $M_2$ |
| $(\mathbf{w}_a, 3)$ | $\{+,-,+,-\}$ | $\Gamma_3$ | $X_3$ | $Y_3$ | $M_3$ |
| $(\mathbf{w}_b, 0)$ | $\{+,-,+,+\}$ | $\Gamma_0$ | $X_3$ | $Y_0$ | $M_3$ |
| $(\mathbf{w}_b, 1)$ | $\{-,+,-,-\}$ | $\Gamma_1$ | $X_2$ | $Y_1$ | $M_2$ |
| $(\mathbf{w}_b, 2)$ | $\{-,-,-,+\}$ | $\Gamma_2$ | $X_1$ | $Y_2$ | $M_1$ |
| $(\mathbf{w}_b, 3)$ | $\{+,+,+,-\}$ | $\Gamma_3$ | $X_0$ | $Y_3$ | $M_0$ |
| $(\mathbf{w}_c, 0)$ | $\{+,+,-,+\}$ | $\Gamma_0$ | $X_0$ | $Y_2$ | $M_2$ |
| $(\mathbf{w}_c, 1)$ | $\{-,-,+,-\}$ | $\Gamma_1$ | $X_1$ | $Y_3$ | $M_3$ |
| $(\mathbf{w}_c, 2)$ | $\{-,+,+,+\}$ | $\Gamma_2$ | $X_2$ | $Y_0$ | $M_0$ |
| $(\mathbf{w}_c, 3)$ | $\{+,-,-,-\}$ | $\Gamma_3$ | $X_3$ | $Y_1$ | $M_1$ |
| $(\mathbf{w}_d, 0)$ | $\{+,-,-,+\}$ | $\Gamma_0$ | $X_3$ | $Y_2$ | $M_1$ |
| $(\mathbf{w}_d, 1)$ | $\{-,+,+,-\}$ | $\Gamma_1$ | $X_2$ | $Y_3$ | $M_0$ |
| $(\mathbf{w}_d, 2)$ | $\{-,-,+,+\}$ | $\Gamma_2$ | $X_1$ | $Y_0$ | $M_3$ |
| $(\mathbf{w}_d, 3)$ | $\{+,+,-,-\}$ | $\Gamma_3$ | $X_0$ | $Y_1$ | $M_2$ |

## C.2 $p_{mm}$-symmetric atomic insulators

Crystals with $p_{mm}$ symmetry possess high symmetry lines as well as high symmetry points. They are given by $l_1 = (k_x, 0)$, $l_2 = (\pi/a, k_y)$, $l_3 = (k_x, \pi/a)$, and $l_4 = (0, k_y)$, and are left invariant by a subgroup of the full point group: $l_1$ and $l_3$ are unchanged under the operations $\{E, M_y\}$, while $l_2$ and $l_4$ are invariant under the action of $\{E, M_x\}$. Each set constitutes a group of its own, which is isomorphic to the cyclic group $Z_2$. The representations can be labelled "+" or "−", depending on whether they are even or odd under the relevant mirror operation.

We can now again use Eq. (5) to deduce the momentum space irreducible representations of both the high symmetry points and the high symmetry lines in the Brillouin zone from the action of the symmetry operations on Bloch functions with any given real-space symmetry label. Consider, for exanple, the elementary band representation $(\mathbf{w}_b, 0)$, which transforms as:

$$M_x \psi^{(\mathbf{w}_b, 0)}(\mathbf{\Gamma}, \mathbf{r}) = M_y \psi^{(\mathbf{w}_b, 0)}(\mathbf{\Gamma}, \mathbf{r}) = \psi^{(\mathbf{w}_b, 0)}(\mathbf{\Gamma}, \mathbf{r}),$$

$$-M_x \psi^{(\mathbf{w}_b, 0)}(\mathbf{X}, \mathbf{r}) = M_y \psi^{(\mathbf{w}_b, 0)}(\mathbf{X}, \mathbf{r}) = \psi^{(\mathbf{w}_b, 0)}(\mathbf{X}, \mathbf{r}),$$

$$M_x \psi^{(\mathbf{w}_b, 0)}(\mathbf{Y}, \mathbf{r}) = M_y \psi^{(\mathbf{w}_b, 0)}(\mathbf{Y}, \mathbf{r}) = \psi^{(\mathbf{w}_b, 0)}(\mathbf{Y}, \mathbf{r}),$$

$$-M_x \psi^{(\mathbf{w}_b, 0)}(\mathbf{M}, \mathbf{r}) = M_y \psi^{(\mathbf{w}_b, 0)}(\mathbf{M}, \mathbf{r}) = \psi^{(\mathbf{w}_b, 0)}(\mathbf{M}, \mathbf{r}).$$

Since the $C_2$ element of the point group is simply the product of the two mirror operations, the mirror eigenvalues by themselves already determine the irreducible representations at the high symmetry points. For example, the fact that $\psi^{(\mathbf{w}_b, 0)}(\mathbf{\Gamma}, \mathbf{r})$ is even under both mirror operations implies that its irreducible representation at $\mathbf{\Gamma}$ is $\Gamma_0$. Similarly, the irreducible representations at $\mathbf{X}$, $\mathbf{Y}$, and $\mathbf{M}$ are found to be $X_3$, $Y_0$, and $M_3$ respectively.

The irreducible representations associated with the high symmetry lines are induced by

those at the high symmetry points. For example, $l_1$ is the line connecting $\mathbf{\Gamma}$ and $\mathbf{X}$. Since it is mapped onto itself by the operation $M_y$, all Bloch states with momenta on the line transform as irreducible representations of the $Z_2$ group $\{E, M_y\}$. Since the momentum can be adiabatically changed along the line, however, there can be no discontinuous changes of the irreducible representation along the line, and all states on $l_1$ must transform under $M_y$ in the same way. In particular, this includes the end points, $\mathbf{\Gamma}$ and $\mathbf{X}$. Since both of these transform with eigenvalue $+1$ under $M_y$, all states along $l_1$ must also be even under the mirror operation. Continuing this way, Table 12 can be constructed, which lists the irreducible representations associated with all high symmetry lines and points in momentum space for each of the possible elementary band representations associated with a $pmm$-symmetric atomic insulator.

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
