# Peer review of "Topological states between inversion symmetric atomic insulators"

_SciPost Physics, doi:SciPost Phys. 10, 137 (2021)_

## Round 1 · Referee Report · Anonymous · 2021-1-22

Strengths
1) this is a very well-written paper
2) the splitting into main text/appendices is appropriate, and the latter provide many useful details on the authors' results
3) the results are general and apply in a model-independent way
4) I believe this work will lead to multiple follow-up theoretical and experimental investigations
Weaknesses
1) it would really help the readers to see one simple example Hamiltonian showing the behavior discussed in the paper
2) I'm confused about the use of the term "topological protection." It might be helpful to better clarify this term.
Report
The authors consider interface states between 2D atomic insulators in class A, which the authors define as systems with vanishing Chern number. Focusing on space groups p2, p3, and pmm, the authors show a simple and generic criterion predicting localized interface states. This is based on the logarithmic derivative of the wavefunction, which is constrained by bulk topology, namely by the symmetry labels of the bands.
I really enjoyed reading this paper, it is very well written. Results are presented in a step-by-step pedagogical way, making the authors' discussion easy to follow also for non-specialists. I think this work is timely and of interest to the community working on topological systems, and it will motivate experimental studies on these kinds of interface states, possibly in various meta-material platforms.
I only have two concerns with regard to this work, and I would like to ask the authors to please address them before I can recommend publication. I recognize that I am biased in this regard, since I have already read the question asked by Daniel Varjas on the SciPost submission page, and I fully agree with him.
The two points are:
1) Please add an example. I understand that the work is general, and that the results apply in a model-independent way: this is one of the strengths of the paper. However, it would really help the readers to have something concrete to point to while going through these general results. With even a simple example in an appendix somewhere, I believe the quality of this paper would be greatly improved.
2) Several times throughout the paper, but especially at the beginning of Section 4, the authors refer to interface states which should be present at the Fermi level. At the end of Section 5, the authors say that these interface states may sensitively depend on crystal terminations. This makes me confused with respect to the way in which the authors define "topological protection."
My confusion is as follows:
As far as I can understand (again, here an example would go a long way), the symmetry labels force interface-bound states to exist, but they don't force these states to continuously interpolate in energy between the valence and the conduction band. My naive guess (which could be wrong) is that these states exist at some particular energy, $E_p$, which is in general different from 0 since there is no chiral or particle-hole symmetry in class A. The interface between two inversion-symmetric bulks is in general not inversion-symmetric, so it might be possible to change the microscopic details of the interface without worrying about inversion breaking. In this case, isn't it possible for me to freely move these interface states in energy, simply by changing the microscopic details of the interface itself? For instance, could I add a chemical potential just to the interface, and push the interface states up in energy until they overlap with the bulk bands?
Note that this would be very different from the "topological protection" of strong TIs (e.g. quantum spin-Hall effect) and what people used to call weak TIs (e.g. 3D stack of 2D QSHE). In those systems, the boundaries are invariant under the symmetries (time reversal, translation) and the boundary states exist at the Fermi level no matter how one changes the microscopic details of the interface, provided its symmetries are left intact.
Are the states discussed in this paper and their amount of topological protection different from Shockley states? Does this paper discuss the conditions for which Shockley states exist in p2, p3, and pmm?
The authors should clarify their definition of topological protection when it comes to the constraints on the energy of interface states. Also they should clarify the connection/difference between the interface states they consider and conventional Shockley states. For instance, with a toy-model example it could be possible to show the states and see if/how they move in energy away from the Fermi level when a chemical potential is added only to the interface region.
Requested changes
See report.
Author: Jasper van Wezel on 2021-05-03 [id 1401]
(in reply to Report 1 on 2021-01-22)Please see attached for our replies to both referees.
Daniel Varjas on 2021-01-08 [id 1129]
It would be very useful if the authors supported their findings with at least one concrete example. In its current form the manuscript only presents very general but somewhat counterintuitive claims. Could the authors show a specific system (given by its Hamiltonian and the action of the symmetries) demonstrating topological interface modes at a heterojunction of two atomic insulators? It would also be useful to elucidate whether the authors mean the term "atomic insulator" as a system with trivial strong invariant, or as a trivial atomic system with respect to the space group symmetries as well.
Author: Jasper van Wezel on 2021-01-13 [id 1148]
(in reply to Daniel Varjas on 2021-01-08 [id 1129])Dear dr. Varjas,
thank you for the helpful comments, which we greatly appreciate.
We tried to explain in the document that 'atomic insulator' is intended here to refer to a material with simple bands that have Chern number zero, without further restrictions. We will try to formulate this more clearly.
We also appreciate your suggestion of including a simulation for an example heterojunction. In fact, we purposely chose not to do that in the current manuscript, because we would like to present a general symmetry-based argument that can be broadly applied. In that light, we feel that a detailed analysis of what must necessarily be an arbitrarily chosen example, is beyond the scope of the current article.
However, if you are interested, we would be very happy to collaborate with you, and look for potentially interesting implementations of the ideas presented here.

---

## Round 1 · Referee Report · Anonymous · 2021-1-27

Report
The authors claim that a group of weak topological invariants, which depend only by the symmetries of the atomic lattice, induces a bulk-boundary correspondence.
In particular, it is claimed that (i) these weak topological invariants predict the presence or absence of states localised at the interface between two inversion-symmetric band insulators with trivial values for their strong invariants, and (ii) the interface modes are protected by the combination of band topology and symmetry of the interface.
These statements are strong and counterintuitive, and I don’t find the supporting arguments in the manuscript convincing. The essential counterargument opposing these statements that it seems likely that it is possible to introduce a perturbation which obeys the symmetries but moves the interface states in energy to the conduction or valence band on either side of the interface. Thus, the interface states hybridise with the bulk states and are no longer localised at the interface. (Notice that the energy gap can be very small on one side of the interface, and therefore intuitively it seems that it would be very easy to hybridize the interface states with the bulk bands on that side of the interface.) This would mean that the interface states are not protected by the combination of the band topology and symmetry of the interface, and therefore also the weak topological invariants would not always predict the presence or absence of localised states at the interface.
It is probably possible to formulate a weaker statement, which is related to statement (i). Namely, I expect that quite generically these weak topological invariants are related to the presence or absence of localised states at the interface. The appearance of such interface states can be understood by first considering a smooth interface with slowly varying parameters connecting the two Hamiltonians, and then realising that one needs quite large perturbation to remove the resulting localised interface states appearing inside the bulk gap. Thus, the interface states will often survive in specific models even in the case of a sharp interface. Nevertheless, I want to emphasise that these localised interface states are not protected just by symmetry and topology; Their appearance requires additional assumptions about the model Hamiltonian.
I am willing to reconsider the paper for publication if the authors have additional arguments to support their statements that the interface states are indeed protected by the combination of band topology and symmetry of the interface. For this purpose I suggest that the authors construct an explicit example (as suggested also both by Daniel Varjas and the first referee), and study the robustness of the interface states with respect to introducing all possible symmetry-preserving perturbations which do not change the band topology.
Author: Jasper van Wezel on 2021-05-03 [id 1402]
(in reply to Report 2 on 2021-01-27)Please see attached for our replies to both referees.
Attachment:

---

## Round 2 · Referee Report · Anonymous · 2021-5-15

Report

The authors have included new examples which nicely illustrate the properties of the interface states.

In my previous report I pointed out that it seems likely that it is possible to introduce a perturbation which obeys the symmetries but moves the interface states in energy to the conduction or valence band on either side of the interface. Thus, the interface states hybridise with the bulk states and are no longer localised at the interface.

In the examples considered by the authors, they indeed found that the interface states can be moved arbitrary close to the gap edge. The authors claim that the topological interface states “enjoy some form of topological protection in the sense that they must exist in the gap and cannot be destroyed or brought into the bulk”. However, one could alternatively argue that since their energy can be brought arbitrarily close to the gap edge also the localization length can diverge, and therefore it is impossible to distinguish them from the bulk states in practice. In my opinion, this is still not discussed clearly enough in the manuscript. The authors call this result counterintuitive and therefore they do not discuss this in the manuscript. I think that this information should be available for the readers of the manuscript. It is therefore necessary to include in the manuscript the figure which the authors have already included in the response letter, and to write it clearly that the interface states can be brought arbitrarily close to the gap edge so that their localization length can diverge.

  • validity: -
  • significance: -
  • originality: -
  • clarity: -
  • formatting: -
  • grammar: -

Author:  Jasper van Wezel  on 2021-05-18  [id 1429]

(in reply to Report 1 on 2021-05-15)

Dear referee,

thank you very much for your report. Making good use of the opportunities provided by the SciPost platform, we would like to already respond to some of your comments, and to ask you to please reply to the message below before we prepare a revised version of the manuscript.

In response to your report, we would like to point out a couple of factual mistakes:

1) You argue that the figure in our reply letter shows that adding an impurity potential along the entire interface allows edge states to be delocalized and absorbed into the bulk, following the intuition you stated earlier. Attractive as your simple intuitive interpretation sounds, however, it is not supported by any of the actual results. The interface states in our figure are exponentially localised, in spite of the impurity strength being absurdly large (i.e. greater than the bandwidth). They do not delocalise and merge with the bulk.

Perhaps the interface states could delocalise and cease to exist in the limit of infinite impurity strenght, but besides that limit not being physical, it also does not apply to our heterojunction setup, as in that limit the two halves of the system will decouple, and not form a heterojunction anymore. The seemingly intuitive picture of being able to push the in-gap band into the bulk thus simply does not apply to the topological heterojunctions considered here, and the in-gap states exist for any non-infinite impurity strength, as predicted in our manuscript.

2) We do not call the fact that states can be moved in energy counterintuitive, and we certainly do not attempt to hide anything whatsoever. In fact, we did explicitly mention the possibility to change the energy of the impurity band in our resubmitted manuscript. The term "counterintuitive" in our previous reply refers to the interface band being split into two parts, one at high energy and localised primarily on one side of the interface, and another at low energy and localised primarily at the other side of the interface. This splitting of the band is counterintuitive in our opinion, and not explained by any physics described in the current manuscript.

It is also particular to the specific setup considered in this specific example suggested by you. It is in no way general to the family of heterojunctions we discuss in the manuscript. This is the reason we left it out of the revised manuscript -- not because we want to hide anything, but because the figure shows a physical effect that is not general and does not apply to most of the systems that this manuscript focusses on. In fact, for as far as we know it might be particular only to the one unphysical setup (with a stretched-out super-strong impurity) that we considered in our reply letter only because you specifically asked for it.

We would like to stress once more that we are not hiding anything. We honestly believe that including an example that shows particular and non-general behaviour would not benefit the readers of our paper. We would also like to stress that the intuition you are arguing the figure should show, is in fact not in agreement with the calculated results.

We would therefore like to ask you, the referee, to please reconsider your recommendation that we include this figure in the manuscript.

If you could please reply to us on the SciPost platform, we can take your reply into account before we formulate any further revisions.

With the best regards, Ana Silva and Jasper van Wezel.

Anonymous on 2021-05-29  [id 1476]

(in reply to Jasper van Wezel on 2021-05-18 [id 1429])

The authors have made a careful analysis of the robustness of the interface states. Moreover, they have given reasonable arguments for not including the figure in the manuscript. Therefore, I will not insist them to include it. I recommend to accept the manuscript for publication in SciPost.

Author:  Jasper van Wezel  on 2021-05-21  [id 1449]

(in reply to Jasper van Wezel on 2021-05-18 [id 1429])

Dear referee,

thank you for your response and for engaging in this discussion before we resubmit. We very much appreciate it.

Thank you also for insisting that we consider how the setup with a stretched-out impurity along the interface evolves as the impurity strength approaches infinity. Doing this allowed us to understand the origin of the split interface states that were mentioned in our previous reply. Although non-generic and not directly related to the subject of the current manuscript (as explained below), we were happy to find that it does bring to the fore a previously unnoticed signature of the topological nature of the interface states.

Since the question of whether the interface states were truly topological was the original reason for you proposing this setup, please allow us to explain. As a very strong impurity potential (exceeding the bandwidth) is added along the entire interface between the two topologically distinct insulators (coupled SSH chains), four things happen simultaneously.

1) the interface state moves away from the interface sites with the impurity potential on them, and instead localises to the left of the row of impurity sites (see right panels of the attached figure 1). At the same time, a different state emerges from the lower bulk band. This state is localised to the right of the impurity site, and its origin is discussed further below. That the original interface state relocates to avoid the sites with an added impurity potential can be interpreted as it flowing around the defect, in direct analogy to the edge states in conventional topological insulators flowing around imperfections in their edges. It is a testament to its topological nature that the interface states is not destroyed by the impurity potential, but merely relocated.

This is further confirmed by the second effect that occurs: 2) a band of impurity states, localised strictly on the impurity sites, is formed at very high energy (the red states in the attached figure 1). These states are disconnected from the bulk materials on either side of the interface, and their energy is determined by the impurity potential. That is, the energy of these states is directly proportional to the impurity potential, and changing the value of the impurity potential moves these states up or down in energy. This is the intuition you referred to in your earlier reports: that a potential at the interface may arbitrarily change the energy of states localised on sites affected by the potential. This intuition is correct, but the affected states are not the original topological states (which relocated to avoid the impurity sites), but rather the non-topological impurity states created by introducing a row of local impurity potentials that makes the affected sites qualitatively different from the surrounding materials.

Although the original topological states do not move in energy proportional to the impurity potential, they are affected by the presence of the impurity potential, in the following way: 3) the row of impurity potentials effectively separates the two halves of the original heterojunction setup. That is, in the presence of a strong impurity potential, the setup is better thought of as a tri-layer setup, with two bulk materials consisting of stacks of SSH chains, separated by an intermediate "spacer" material. In the limit of the impurity potential going to infinity, these three layers decouple entirely. Evolving the impurity strength from zero to infinity thus corresponds to smoothly interpolating between a two-layer-interface setup, and a setup of three isolated materials with edges. The final result is not specific to the original interface, but instead dominated completely by the properties of the individual isolated materials. This can be clearly seen by following the two states moving through the gap. One of those (localised on the right and emerging from the lowest bulk band as the impurity potential grows), evolves towards an edge state of the two-band material on the right. The other (the relocated original interface state) will merge with the highest bulk band and become a bulk state of the material on the left in the limit of infinite impurity strength. We would like to stress again that this behaviour is due to the properties of the final setup of separated bulk materials, and not the initial heterojunction. By choosing a different form or location for the row of impurities, we could have easily constructed a situation in which there are two edge states at infinite impurity strength, or zero. The evolution of the interface state into either an edge or bulk state is thus not generic to the heterojunction setup that is the subject of the current manuscript. It is determined entirely by the choice of the final setup of three disconnected materials, which is not described by the analysis in the current manuscript.

Having said this, it is interesting to see how the original interface state transfroms into a bulk state. This is the fourth thing that happens: 4) The relocated interface state moves up in energy as the impurity potential is increased, even though its energy is not directly proportional to the impurity potential. If the bandwidth is (realistically) taken to be a few eV, the relocated interface state moves by less than an eV as the impurity potential is increased by tens of eV (that is, ~ 10^5 Kelvin). Eventually, it approaches the edge of the bulk band (with impurity strength now exceeding twice the bandwidth), and as you correctly predicted, the relocated interface state then starts to hybridize with the bulk states of the left bulk material. The hybridized state is not exponentially localised next to the impurity anymore, but develops a maximum that moves away from the interface and into the bulk as the impurity potential is further increased (see the attached figure 2). The hybridized state is still more localised towards the interface than towards the centre of the bulk for any finite impurity potential, and thus retains a component of the localized interface state. Only at truly infinite impurity potential, when the setup has no remnant at all of the original heterojunction setup, does the state become completely bulk.

We believe all of the above analysis strengthens the two main arguments of our previous reply: - The heterojunction interface states are truly topological. This is confirmed by their relocating away from the impurity potential, and by their energy not being proportional to the impurity potential. - The setup proposed by the referee, with an unrealistically strong impurity potential (tens of thousands of Kelvins) stretched over unrealistically long distances (the entire interface, which could be centimetres long), does not show behaviour that is generic for the heterojunction setup which is the subject of the current manuscript. Instead, its physics is entirely dominated by the limit of three disconnected materials, which is not treated at all in the current manuscript, and which is qualitatively different for different choices of impurity potential.

We therefore insist that our original conclusion about the topological nature of the interface states is correct, and we maintain our conviction that the discussion and figure of a stretched impurity potential evolving into a three-material-setup do not belong in the current manuscript. They are presented in this reply on the SciPost platform, which will be forever and freely accessible to interested readers, and we believe that suffices.

However, if you (as referee) insist that we must include the above discussion in the manuscript, despite our objection that it will be confusing to readers because it addresses the physics of a setup that is not the topic of the manuscript, we are of course at your mercy. If this allows you to support publication, we will include it. But as said before, we actually hope that in light of the analysis above, you will already be able to support publication with the discussion available here, but not in the manuscript. Please let us know.

Best regards, Ana Silva and Jasper van Wezel.

Attachment:

Reply_figure.pdf

Anonymous on 2021-05-18  [id 1430]

(in reply to Jasper van Wezel on 2021-05-18 [id 1429])

The authors write that although the energy of the interface states can be arbitrarily close to the gap edge, the edge states do not delocalize to the bulk. I would be surprised if this is indeed the case. In this case, it is very important that the authors plot the localization length of the edge states as a function of the parameter which controls the interface state energies. Naively, I would expect that the localization length diverges as their energy approaches the gap edge. This would of course mean that the edge states delocalize into the bulk as I suggested. But if this is not the case I will reconsider my report.

I also do not understand how a specific example is not part of a general family of heterojunctions. Also I do not see any reason why the generic case would be the one where the energies are deep inside the band gap. If one for example considers the simple models for topological insulators the Dirac point of the surface states tends to be almost always deep inside the gap. On the other hand, in realistic models it is almost always buried inside the bulk bands. In those systems, the energy of the Dirac point is not very important because the surface state energies always connect the valence and conduction bands. But in the case of the heterojunctions discussed in the manuscript the energies of the interface states are very important. If there is no reason why they should be deep inside the gap, isn't the general case the one where they are not?

---

## Round 2 · Author Response

Dear editor,

We would like to thank the referees for their time and effort in reviewing our submission, and for their many positive comments regarding our work. In particular, the referees note that the paper is “very well written", presents "strong" or "general, model-independent" results, and is likely to "lead to multiple follow-up theoretical and experimental investigations".

Both referees raised similar concerns, which we believe are primarily based in our failure to give a sufficiently clear definition of our use of the terms "strong topology" and "topological protection" in the original manuscript. Additionally, both referees ask for an explicit example of our proposed interface states.

In the attached, revised manuscript we address both concerns of the referees in detail. We extended the introduction, and now explicitly define the meaning of all terms that are open to multiple interpretations. We also introduce an extensive new section with three additional figures, describing two explicit examples in which the proposed topological interface states arise. We show their existence in numerical simulations, and probe their robustness in the way suggested by the referees.

We believe these changes adequately address the concerns raised by the referees, and respectfully ask you to reconsider the attached manuscript. Below, we respond to the comments of both referees point by point.

Yours sincerely,

Ana Silva and Jasper van Wezel.

---

## Round 2 · List of Changes

Our reply letter includes a figure, which we cannot upload here -- For our replies to the comments made by referees, and an explanation of the changes made, please see our "response to the referee comments" (submitted earlier to the SciPost page for this submission).

---

## Editorial Decision

published